# Mutual Information Dynamics Learning: A New Unsupervised Reinforcement Learning Paradigm

## Abstract

Unsupervised reinforcement learning (URL) aims to develop general-purpose agents that can adapt to unseen downstream tasks without relying on task-specific supervision. Existing approaches predominantly focus on learning diverse skills by maximizing mutual information, but they are often limited to simple navigation tasks and fail to scale to more complex domains such as robotic manipulation, where prior knowledge is typically required. In this work, we demonstrate that mutual information-based objectives can be leveraged far beyond skill learning. We propose a novel URL framework that trains exploratory skills to collect diverse transition data with distinct dynamics. This diverse dataset enables the training of a mixture of dynamic models, where each model captures the dynamics of a specific region. Collectively, these models provide comprehensive coverage of the dynamics required for a wide range of downstream tasks. Our straightforward and prior-free learning objective outperforms existing state-of-the-art skill discovery approaches in URL. Our results advocate a paradigm shift in URL, from skill learning toward dynamics learning, to acquire fully generalizable knowledge during pretraining.

## 1 Introduction

After the success of Reinforcement Learning (RL) in domains such as autonomous control (Kiumarsi et al., 2017), Go (Silver et al., 2016), and video games (Mnih et al., 2013; Vinyals et al., 2019), its widespread adoption has been hindered by a fundamental limitation: high sample complexity. Motivated by the pretrain–finetune paradigm that has driven progress in natural language processing (Radford et al., 2019; Devlin et al., 2019) and computer vision (Henaff, 2020; He et al., 2020), a new research direction has emerged: Unsupervised Reinforcement Learning (URL). While pretraining in RL also includes representation learning (Touati & Ollivier, 2021; Wu et al., 2018), in this work we specifically use URL to refer to unsupervised behavior learning where agents develop exploratory behaviors and acquire potentially useful primitive skills. In this setting, agents are pretrained without access to task-specific rewards and instead rely on intrinsic motivations (Oudeyer & Kaplan, 2009). These intrinsic signals encourage agents to discover a diverse repertoire of skills, which can later be rapidly adapted to novel downstream tasks.

A common approach for unsupervised skill learning of URL is Mutual Information Skill Learning (MISL) that its pretraining aim to maximize the mutual information between state and skill latent (Eysenbach et al., 2019b; Florensa et al., 2017; Hansen et al., 2020; Liu & Abbeel, 2021b). The intuition is that by maximizing this mutual information the choice of skills can effectively affect where the states are distributed so that these skills could be used for tasks like navigation to certain regions.

However, basic MISL objectives struggle to capture complex behaviors, making them inefficient for tasks such as game playing and robotic manipulation. Various extensions to MISL have been proposed, but even the most recent works (Park et al., 2024b; Zheng et al., 2025), despite receiving significant attention from the community, remain largely limited to goal-reaching downstream tasks. To address this limitation and enable the discovery of more diverse and intricate skills, researchers have incorporated additional prior knowledge into the MISL objective. For instance, DADS (Sharma et al., 2020) integrates knowledge about the relevance of specific state dimensions

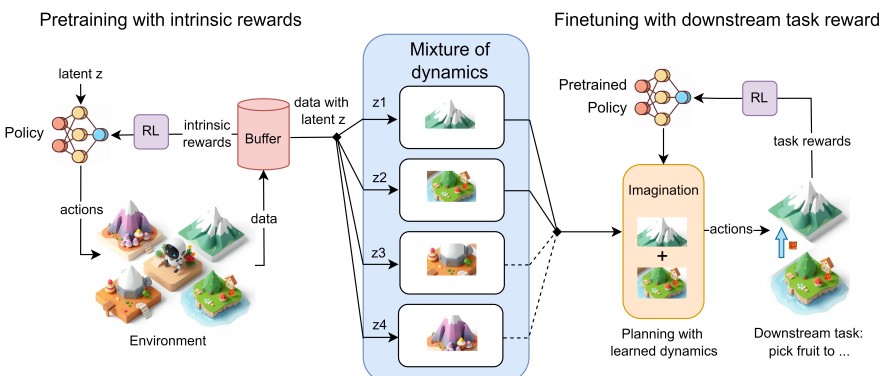

Figure 1: **Overview of MIDL:** In the pretraining stage, we update the policy using intrinsic rewards and collect data through environment interactions to pretrain the dynamic models. This phase focuses on learning a diverse and accurate ensemble of dynamic models while encouraging exploration. In the finetuning stage, we leverage the pretrained dynamic model ensemble for planning, guided by the pretrained policy, to enable fast adaptation to downstream tasks.

for robotic locomotion, while VISR (Hansen et al., 2020) and APS (Liu & Abbeel, 2021b) assume linear reward features for downstream tasks, thereby allowing MISL to learn transferable successor features (Barreto et al., 2017) applicable to game playing. Although these approaches extend the applicability of MISL, they still show unsatisfactory fine-tuning performance on standard robotic manipulation benchmarks (Tunyasuvunakool et al., 2020; Todorov et al., 2012). Results from a widely used robotic control benchmark for URL (Laskin et al., 2021) demonstrate that MISL algorithms such as DIAYN (Eysenbach et al., 2019b), SMM (Lee et al., 2019), and APS (Liu & Abbeel, 2021b) achieve significantly worse downstream performance compared to pure exploration methods (Liu & Abbeel, 2021a; Pathak et al., 2019) that do not involve skill learning. More recent work, CIC (Laskin et al., 2022), was introduced as a MISL method, but in practice its pure exploration variant achieves stronger performance on this benchmark than its skill-learning counterpart. This disappointing outcome, showing that current unsupervised skill learning methods do not adequately prepare agents for practical robotic control, raises an important question for future URL research:

*Should we continue refining MISL objectives with increasingly complex modifications and additional prior knowledge? Or should we instead investigate what else agents can learn during unsupervised pretraining besides skills?*

Our answer is the latter. We maintain a simple mutual information objective with minimal prior assumptions, and shift the focus from skill learning to dynamics learning. Specifically, we learn basic skills that target distinct regions of the state space, thereby encouraging diversity in the discovered areas. Each region exhibits unique dynamics, yielding transition datasets with diverse dynamics. From these datasets, we train an ensemble of dynamics models, where each model specializes in a particular region, collectively covering the full range of explored dynamics (see Figure 1). We present this framework as a new paradigm for URL: **Mutual Information Dynamics Learning (MIDL)**.

While model-based URL methods such as P2E (Sekar et al., 2020) and UMP (Rajeswar et al., 2022) have been proposed, they rely on model-free intrinsic rewards (Pathak et al., 2019; Mazzaglia et al., 2022) without incorporating skill learning or explicitly tailoring their objectives for model-based pretraining. EUCLID (Yuan et al., 2023) moves closer by training an ensemble of dynamics models on skill-conditioned data, yet the diversity in its action space does not

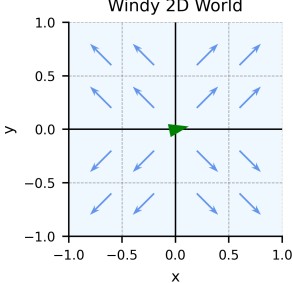

Figure 2: Toy 2D environment with quadrant-dependent wind. The green triangle is the agent. MIDL learns all 4 distinct wind dynamics via an ensemble of 4 models, while previous model-based URL methods can not, as shown in Table 6.

guarantee diversity in the learned environmental dynamics. To highlight this limitation, we consider a simple 2D continuous point-mass environment with quadrant-dependent wind dynamics (Figure 2), where $s' = s + a + wind$. In this setting, only our proposed MIDL successfully recovers all four distinct dynamics using an ensemble of four specialized models, whereas both UMP and EUCLID fail to capture the complete dynamics structure.

We summarize our contributions as follows:

- We show that increasingly complex, prior-driven skill learning does not necessarily improve downstream performance in URL. Instead, we advocate shifting the focus from unsupervised skill learning to unsupervised dynamics learning, which provides general knowledge beneficial across diverse downstream tasks.

- We propose an elegant model-based URL framework that employs a mutual information objective with minimal prior assumptions. The framework leverages skill discovery to collect data from regions with distinct dynamics, enabling the construction of a dynamics ensemble in which each model specializes in a particular region. This design provides a simple yet intricate integration of MISL with model-based pretraining.

- Through extensive experiments, we demonstrate that our approach learns more diverse and robust environmental dynamics than existing model-based methods. Furthermore, the learned dynamics lead to state-of-the-art performance on the URL benchmark and a challenging Humanoid environment, while remaining compatible with most model-based backbone RL algorithms.

## 2 PRELIMINARIES

**MDP without external rewards**   We consider infinite-horizon MDPs $\mathcal{M} = (\mathcal{S}, \mathcal{A}, P, p_0, \gamma)$ *without external rewards* with discrete states $\mathcal{S}$ and actions $\mathcal{A}$, dynamics $P(S_{t+1}|, S_t, A_t)$, initial state distribution $p_0(S_0)$, and discount factor $\gamma \in [0, 1]$. We denote capital $S$ as the Random Variable (RV) of state and the lowercase $s$ as a sample of state. A policy $\pi(A|S)$ has its discounted state occupancy measure as $p^\pi(S) = (1 - \gamma) \sum_{t=0}^{\infty} \gamma^t P_t^\pi(S)$, where $P_t^\pi(S)$ is the probability that policy $\pi$ visits state $s$ at time $t$. There can be downstream tasks that define extrinsic reward as a state-dependent function $r(s)$, where action-dependent reward functions can be handled by modifying the state to include the previous action. The cumulative reward of the corresponding downstream task is $\mathbb{E}_{s \sim p^\pi(S)}[r(s)]$.

The problem of unsupervised skill learning is to learn a skill-conditioned policy $\pi(A|S, z)$ where $z \in \mathcal{Z}$ represents the latent skill. The skill latent space $\mathcal{Z}$ can be either continuous $\mathbb{R}^d$ or discrete $\{z_1, z_2, ..., z_{N_z}\}$. $H(\cdot)$ and $I(\cdot; \cdot)$ denote entropy and mutual information, respectively.

**Unsupervised Reinforcement Learning (URL)**   Without external rewards, Unsupervised Reinforcement Learning (URL) typically focuses on exploration and skill learning.

For exploration, intrinsic motivation is often based on curiosity or surprise in environmental dynamics, as in ICM (Pathak et al., 2017), RND Burda et al. (2019) and Disagreement Pathak et al. (2019). Another approach is to maximize state entropy. For example, SMM Lee et al. (2019), APT (Liu & Abbeel, 2021a), ProtoRL (Yarats et al., 2021), and APS (Liu & Abbeel, 2021b).

Beyond exploration, a key goal of URL is unsupervised skill learning. A common approach is Mutual Information-based Skill Learning (MISL), used in methods like VIC (Gregor et al., 2017), DIAYN (Eysenbach et al., 2019a), and VALOR (Achiam et al., 2018). The MISL objective, also known as empowerment Salge et al. (2014), seeks to learn a skill latent conditioned policy $\pi(A|S, Z)$ by maximizing the mutual information between states and the skill latent variable:

$$\max_\pi I(S; Z) = H(S) - H(S|Z) \tag{1}$$

This objective encourages high overall state entropy across skills while minimizing entropy within each skill, effectively partitioning the state space into distinct regions. As a result, the agent learns diverse, fundamental skills useful for tasks like navigation and locomotion, as shown in Campos et al. (2020); Sharma et al. (2020). Theoretical work Eysenbach et al. (2022); Yang et al. (2024) also explores how MISL-learned skills can be adapted to downstream tasks. However, for more complex

tasks such as robotic control (Tunyasuvunakool et al., 2020) and video game playing (Bellemare et al., 2013), MISL often underperforms compared to pure exploration methods like RND, which do not involve skill learning, as shown in Figure 3. This raises the question of how to improve MISL so its learned skills are more beneficial for complex downstream tasks.

DADS Sharma et al. (2020) incorporates prior knowledge about key state dimensions, enabling skills to focus on diverse behaviors along those axes. VISR Hansen et al. (2020) introduces successor features (Barreto et al., 2018) into skill learning by assuming downstream rewards can be linearly represented. It approximates mutual information as an expectation over a linear function, allowing fast adaptation via linear regression. APS (Liu & Abbeel, 2021b) builds on VISR and performs well in Atari (Bellemare et al., 2013), but is still short of pure exploration methods in robotic control. CIC (Laskin et al., 2022) takes a different route by optimizing mutual information over transitions using contrastive learning, but still underperforms compared to simply maximizing transition entropy without skill learning.

**What's Next for URL**  The tasks in Laskin et al. (2021), though more complex than basic navigation, remain relatively standard within reinforcement learning. Real-world applications are expected to be far more challenging Dulac-Arnold et al. (2021); Polydoros & Nalpantidis (2017). Since URL aims to pretrain agents with general, transferable knowledge, it must scale to larger, more complex state spaces. As these spaces grow, the likelihood of learning a skill closely aligned with a specific downstream task decreases, highlighting the limitations of mutual information skill learning (MISL) and motivating a rethink of the current URL paradigm.

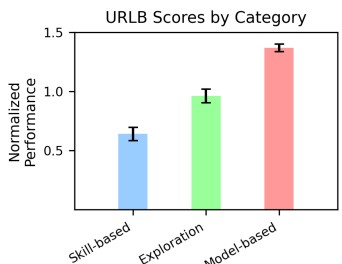

Figure 3: Average downstream task performance scores on the URL benchmark (Laskin et al., 2021) based on dm-control (Tunyasuvunakool et al., 2020), normalized over the performance of CIC. Skill-based methods (DIAYN, APS), which optimize the MISL, exhibit inferior downstream task performance compared to exploration-based methods (Disagreement, CIC) that do not learn explicit skills. Model-based methods (UMP, EUCLID, MIDL (ours)) significantly outperform the aforementioned model-free approaches.

Rather than relying on increasingly tailored skill learning with added assumptions, we advocate focusing on efficient dynamics learning. A strong understanding of environmental dynamics enhances generalization and adaptability, making URL more applicable to real-world scenarios. Empirical studies (Sekar et al., 2020; Rajeswar et al., 2022) combining model-based RL (Hafner et al., 2020b) with pure exploration rewards (Pathak et al., 2019; Mazzaglia et al., 2022), an approach we term Unsupervised Model-based Pretraining (UMP), have demonstrated superior performance in robotic tasks (Laskin et al., 2021), underscoring the promise of dynamics-focused URL.

Rather than discarding MISL, we integrate it with model-based URL to guide agents toward regions with distinct dynamics. By optimizing a mutual information objective, our approach promotes skill discovery that generates diverse state transitions, enabling the training of an ensemble of specialized dynamics models. Each model focuses on a unique region, and collectively the ensemble captures the full spectrum of environmental dynamics. We call this framework Mutual Information Dynamics Learning (MIDL).

## 3  METHOD

In this work, we propose MIDL, which is a mutual information-based URL method for model-based pretraining that seamlessly bridges the gap between MISL and model-based URL. As illustrated in Figure 1, our objective is to train a dynamic ensemble during the pretraining phase and subsequently utilize it for planning in downstream tasks. This approach is characterized by the following key properties:

- Our intrinsic reward guides the agent to discover regions with diverse dynamics and collect transition samples from those regions.
- Each model in the ensemble specializes in the distinct dynamics of its corresponding region.

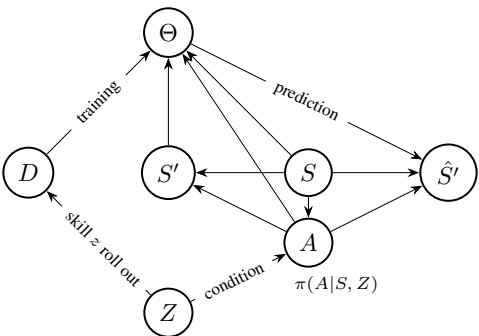

Figure 4: Probabilistic graphical model of our setting, $Z$ is RV for the conditional input of the policy, $D$ is RV for the collected transitions in the buffer, $\Theta$ is the RV for the dynamic model parameters. $S'$ is the next state by the true dynamic of the environment. $\hat{S}'$ is predicted by dynamic model $\Theta$ with current state $S$ and current action $A$ as inputs. We can see from the graph that dynamic model $\Theta$ is dependent on its training data $D$ and conditional input $Z$, since policy with different conditional input $Z$ collects different data $D$.

- It selects suitable dynamic models for trajectory planning in downstream tasks.

MIDL not only enhances the downstream task adaptation ability of URL, but also importantly demonstrates how the findings from mainstream MISL research can be effectively transferred to facilitate efficient dynamics learning.

## 3.1 OBJECTIVE DESIGN

Similar to MISL, MIDL also formulates the agent's policy as $\pi(A|S, Z)$. We use a probabilistic graph shown in Figure 4 to model the model-based URL learning procedure, where $D$ is RV for the collected transitions in the buffer and $\Theta$ is the RV for the dynamic model parameters. $S'$ is the next state of the true dynamic of the environment that is conditioned on current state $S$ and current state $A$. $\hat{S}'$ is the predicted next state that depends on current state $S$, current state $A$, and dynamic $\Theta$. $Z$ is similar to the skill latent of MISL, which is a conditional input of the policy and affects the agent's behavior. Therefore, action $A$ and buffer data $D$ are both dependent on $Z$. Since $Z$ affects the data $D$ in the buffer, $\Theta$ is also dependent on $Z$.

We propose the following mutual information objective for MIDL:

$$
\begin{aligned}
& I(\hat{S}'; (S', Z)|S, A) \\
=\; & I(\hat{S}'; S'|S, A) + I(\hat{S}'; Z|S, A, S') \\
=\; & \underbrace{I(\hat{S}'; S'|S, A)}_{\text{1.Information gain}} + \underbrace{\mathbb{E}\big[\log \frac{p(\hat{s}'|s, a, s', z)}{p(\hat{s}'|s, a, s')}\big]}_{\text{2.Dynamic Matching}}
\end{aligned}
\tag{2}
$$

Intuitively, $I(\hat{S}'; (S', Z)|S, A)$ encourages the next state prediction $\hat{S}'$ to be informative about both the true next state $\hat{S}'$ and skill (ensemble index) $Z$, so that the ensemble captures diverse yet accurate dynamics. The first term represents an *information gain*, quantifying how much is learned about how much information prediction $\hat{S}'$ has about the data $(S, A, S')$. This can be approximated by the prediction error between $S'$ and $\hat{S}'$, like ICM (Pathak et al., 2017) or the variance of predicted $\hat{S}'$, like Disagreement (Shyam et al., 2019; Park & Levine, 2023). Maximizing it promotes exploration in regions where the model is uncertain, helping the agent learn more about the dynamics of the environment. The second term, *dynamic matching*, encourages the $Z$-conditioned policy to focus on regions where its corresponding dynamic model predicts better than the ensemble average, i.e., where $p(\hat{s}'|s, a, s', z)$ is higher than $p(\hat{s}'|s, a, s')$. This allows the model $z$ to specialize in the data collected by skill $z$.

By combining these two objectives, the agent learns a policy that not only explores effectively but also discovers diverse dynamics while collecting data to improve the overall model ensemble. This

approach ensures a balance between exploration and specialization, enabling the agent to adapt to varying dynamics in the environment.

## 3.2 Practical Implementation

The objective in Equation (2) can be further expanded.

$$I(\hat{S}'; (S', Z)|S, A) \tag{3}$$

$$= I(\hat{S}'; S'|S, A) + \mathbb{E}\big[\log \frac{p(\hat{s}'|s, a, s', z)}{p(\hat{s}'|s, a, s')}\big] \tag{4}$$

$$\approx \underbrace{I(\hat{S}'; S'|S, A)}_{\text{1.Information gain}} + \underbrace{\mathbb{E}_{z \sim p(z)}\mathbb{E}_{(s,a,s') \sim \pi_\phi^z}\big[\log \frac{q_\theta(s'|s, a, z)}{q_{\theta,\psi}(s'|s, a)}\big]}_{\text{2.Dynamic Matching}} \tag{5}$$

where $q_\theta(s' \mid s, a, z)$ denotes the prediction of the $z$-th model, trained using data collected from skill $z$. A detailed derivation from (4) to (5) is provided in Section E.1. The term $q_{\theta,\psi}(s'|s, a)$ denotes the ensemble average prediction, which is

$$q_{\theta,\psi}(s'|s, a) = \sum_z q_\theta(s'|s, a, z)q_\psi(z|s, a) \tag{6}$$

Prior work (Sharma et al., 2020; Park & Levine, 2023) often assumes a uniform prior over skills, i.e., $p(Z|S, A) \approx \frac{1}{|\mathcal{Z}|}$, which is a very coarse approximation. Instead, we train a gating network $q_\psi(z|s, a)$, which enables adaptive weighting of the ensemble models, leveraging the specialization of each. More details are discussed in Section F.

As mentioned, the information gain term can be approximated using a pure exploration signal, such as the prediction error between $S'$ and $\hat{S}'$ (ICM) or prediction variance of $\hat{S}'$ (Diagreement) to estimate it. Further details are provided in Section E. When using Disagreement for the exploration signal, we have

$$I(\hat{S}'; (S', Z)|S, A) \approx \underbrace{\mathbb{E}_{s,a}\left[\text{Var}\left(q_\theta(s, a)\right)\right]}_{\text{1.Information gain}} + \underbrace{\mathbb{E}_{z \sim p(z)}\mathbb{E}_{(s,a,s') \sim \pi_\phi^z}\big[\log \frac{q_\theta(s'|s, a, z)}{q_{\theta,\psi}(s'|s, a)}\big]}_{\text{2.Dynamic Matching}} \tag{7}$$

### 3.2.1 Intrinsic reward design

Using the practical approximations above, the intrinsic reward for optimizing Equation (7) with the policy on each transition sample $(s, a, s', z)$ is

$$\log q_\theta(s'|s, a, z) - \log q_\theta(s'|s, a) + \beta \text{ Var}\left(q_\theta(s, a)\right) \tag{8}$$

where $\beta$ denotes a coefficient to balance exploration and dynamic matching.

## 3.3 Learning Model Ensemble

The dynamic matching of the objective guides the skills to reach diverse regions with distinct dynamics. Here we explain how the collected data is used for dynamic model learning.

We denote $\theta$ as the parameter model ensemble, and the $z$-th model has the prediction of $q_\theta(s'|s, a, z)$. Then each model is learned from the data collected by its corresponding skill:

$$\max_\theta \mathbb{E}_z \left[\mathbb{E}_{s,a,s' \sim p(D|z)}\left[\log q_\theta(s'|s, a, z)\right]\right], \tag{9}$$

and learn $\psi$ by

$$\max_\psi \mathbb{E}_{s,a,s' \sim D}\left[\log \sum_z q_\theta(s'|s, a, z)q_\psi(z|s, a)\right]. \tag{10}$$

More details on how to compute Equations (9) and (10) are in Section F. Moreover, there can be alternative ways for data matching, and they are discussed in Section G.

---

**Algorithm 1** Pretraining of MIDL

---

1: **Input:** number of models in ensemble $|\mathcal{Z}|$
2: **Initialize:** Model $\theta = [\theta_1, \theta_2, ..., \theta_{|\mathcal{Z}|}]$, Selector $\psi$, Policy $\pi_\phi$, Dataset $D = [D_1, D_2, ..., D_{|\mathcal{Z}|}]$.
3: **while** pretraining **do**
4:     Sample $z$ uniformly from $\mathcal{Z} \cup \{z'\}$
5:     Rollout with $\pi_\phi^z$ and collect data to $D_Z$.
6:     Update $\theta, \psi$ with Equations (9) and (10)
7:     **if** $z == z'$: **then** Update $\pi_\phi$ with pure exploration intrinsic reward
8:     **else**: Update $\pi_\phi$ with MIDL intrinsic reward Equation (8)
9:     **end if**
10: **end while**

---

**Algorithm 2** Finetuning of MIDL

---

1: **Input:** Model $\theta = [\theta_1, \theta_2, ..., \theta_{|\mathcal{Z}|}]$, Selector $\psi$, Policy $\pi_\phi$,
2: **Initialize:** Dataset $D$.
3: **while** finetuning **do**
4:     Planning with $\pi_\phi^{z'}$ and $\theta$ and collect data to $D$.
5:     Update $\theta$ with data from $D$
6:     Update $\pi_\phi$ with data from $D$ with normal RL updates
7: **end while**

---

### 3.4 ALGORITHM

Like other URL methods, MIDL follows two stages: pretraining and fine-tuning. The pretraining algorithm for MIDL is outlined in Algorithm 1. Since MIDL focuses exclusively on learning policies for dynamic learning, there is no guarantee that these policies will serve as effective initialization for downstream tasks. Inspired by the observation that pure exploration methods often perform best in model-free URL settings, we introduce an additional policy, $\pi_\phi^{z'}$, trained using pure exploration signals such as Disagreement (Pathak et al., 2019), and leverage it for downstream task adaptation. The finetuning algorithm for MIDL is outlined in Algorithm 2. The learned dynamic model ensemble $\theta$ is utilized for planning, and the policy $\pi_\phi^{z'}$ serves as the initialization for the planning process. Subsequently, the policy is updated solely based on the task reward. This approach ensures that the policy is fine-tuned effectively to maximize performance on the specific downstream task while leveraging the pre-trained dynamic models for efficient planning such as MPPI Williams et al. (2015).

## 4 EXPERIMENTS

MIDL provides insight into how to efficiently utilize the mutual information objective, a URL signature, for unsupervised dynamics learning. The experiments in this section aim to address the following key questions: Can MIDL learn diverse dynamics? Can MIDL facilitate the learning of a more accurate model ensemble? Can MIDL enhance the downstream task performance?

Table 1: Learned wind vectors in each model of the ensemble across methods. Only MIDL successfully captures all four distinct quadrant-specific dynamics.

|  | UMP | EUCLID | MIDL (Ours) |
|---|---|---|---|
| Model 1 | +0.2003, -0.1983 | +0.1991, -0.1999 | +0.2003, -0.1991 |
| Model 2 | +0.0675, -0.2738 | +0.1997, +0.1997 | -0.1998, +0.1993 |
| Model 3 | -0.2003, -0.1990 | -0.2000, +0.1993 | -0.1994, -0.1999 |
| Model 4 | -0.1996, -0.1996 | -0.1976, +0.2017 | +0.1991, +0.1996 |

### 4.1 EXAMPLE OF DYNAMIC DIVERSITY

We consider the simple 2D continuous point mass environment with dynamics $s' = s + a + \text{wind}$, illustrated in Figure 2, where each quadrant has a distinct wind vector: $[0.2 \operatorname{sign}(x), 0.2 \operatorname{sign}(y)]$. We run MIDL and two baselines. One is pure exploration baselines such as Sekar et al. (2020); Rajeswar et al. (2022). They share the high-level idea of integrating pure exploration methods for model-based URL, differing primarily in their lower-level implementations, so we categorize them as Unsupervised Model-based Pretraining (UMP) methods that rely on pure exploration intrinsic rewards for learning.

Another baseline is Euclid (Yuan et al., 2023), which trains an ensemble of dynamics models using skill-conditioned data, but relies on the heuristic that diversity in the action space implies diversity in dynamics, an assumption with no guarantees. Each method learns an ensemble of four learnable wind vectors. After 5k unsupervised steps, only MIDL successfully recovers all four quadrant-specific dynamics, while Euclid recovers three and UMP only two. More details are in Section H.2.

## 4.2 MODEL ACCURACY ANALYSIS

To address whether MIDL facilitates a more accurate model ensemble, we compare it to UMP and Euclid. For a fair comparison, we unify the model-based backbone as TDMPC (Hansen et al., 2022), and learn a gating network $q_\psi(z|s,a)$ so that all methods can exlploit the mixture of all models. Specifically, UMP does not incorporate Dreamer as described in Sekar et al. (2020); Rajeswar et al. (2022). Instead, it is TDMPC pretrained with Disagreement (Pathak et al., 2019).

To assess the dynamic learning capabilities of different model-based URL methods, we formulate a two-player competitive game. In this game, one player represents the policy, while the other represents the model. The policy player aims to collect more challenging data that induce higher prediction errors, while the model player strives to improve accuracy by minimizing prediction errors. This adversarial setup provides a rigorous framework for evaluating the robustness and effectiveness of each method in dynamic learning. A pay-off matrix of this game is shown in Table 2.

This table shows the L2 prediction error of the methods evaluated in the Swimmer environment (Brockman et al., 2016). The results demonstrate that MIDL collects the most challenging data while maintaining the highest accuracy across datasets generated by all methods. This empirically validates that MIDL, as a more principled approach, indeed learns a better dynamic model ensemble than previous heuristic methods.

Table 2: L2 prediction error of different models evaluated on data collected by various policies.

| Data\Model | MIDL(Ours) | Euclid | UMP | Difficulty ↑ |
|---|---|---|---|---|
| MIDL | **0.47** | **0.53** | **0.57** | **1.57** |
| Euclid | **0.27** | 0.38 | 0.42 | 1.07 |
| UMP | **0.29** | 0.31 | 0.29 | 0.89 |
| Accuracy ↓ | **1.03** | 1.22 | 1.28 | |

## 4.3 DOWNSTREAM TASK PERFORMANCE

Table 3: Full results of pre-training for 2M and fine-tuning for 100k steps on URLB (Laskin et al., 2021). The results of DIAYN, APS, Disagreement are from Laskin et al. (2021). CIC, UMP and EUCLID results are from Yuan et al. (2023).

| Domain | Task | DIAYN | APS | Disagreement | CIC | UMP | EUCLID | MIDL (Ours) | Oracle |
|---|---|---|---|---|---|---|---|---|---|
| walker | Flip | 381±17 | 461±24 | 491±21 | 631±34 | 971±1 | 969±2 | **973±4** | *975* |
| | Run | 242±11 | 257±27 | 444±21 | 486±25 | 765±10 | 770±9 | **788±16** | *806* |
| | Stand | 860±26 | 835±54 | 907±15 | 959±2 | 985±1 | 985±1 | **986±2** | *991* |
| | Walk | 661±26 | 711±68 | 782±33 | 885±28 | 967±4 | 972±1 | **975±3** | *984* |
| Quadruped | Jump | 578±46 | 538±42 | 668±24 | 595±42 | 840±13 | 858±14 | **864±10** | *881* |
| | Run | 415±28 | 465±37 | 461±12 | 505±47 | 651±35 | 735±16 | **762±70** | *793* |
| | Stand | 706±48 | 714±50 | 840±33 | 761±54 | 953±6 | **958±5** | 953±20 | *980* |
| | Walk | 406±64 | 602±86 | 721±56 | 723±43 | 872±42 | 925±6 | **939±18** | *975* |
| Jaco | Reach bottom left | 17±5 | 96±13 | 134±8 | 138±9 | 214±5 | **220±3** | 213±13 | *225* |
| | Reach bottom right | 31±4 | 93±9 | 122±4 | 145±7 | 205±9 | 212±2 | **217±8** | *223* |
| | Reach top left | 11±3 | 65±10 | 117±14 | 153±7 | 197±23 | **225±5** | 225±7 | *229* |
| | Reach top right | 19±4 | 81±11 | 140±47 | 163±4 | 219±7 | **229±6** | 224±3 | *231* |

It is natural to expect a correlation between the accuracy of the dynamic model and the performance of the downstream tasks, as empirically demonstrated in Rajeswar et al. (2022). To explicitly show that MIDL enhances downstream task performance, we evaluate it in two benchmarks:

**Environments:** We evaluate MIDL in two settings: (1) the standard Unsupervised Reinforcement Learning Benchmark (URLB (Laskin et al., 2021)), and (2) the more realistic, high-dimensional Humanoid environment (Tunyasuvunakool et al., 2020). URLB includes three domains—Walker, Quadruped, and Jaco—each with four challenging downstream tasks (e.g., flip, jump, run). The Humanoid environment includes three tasks: stand, walk, and run. These benchmarks assess MIDL's effectiveness across diverse and increasingly complex control scenarios.

**Baselines:** We compare our method against the following baselines (details in Section D). Recent URL methods such as Park et al. (2024b); Zheng et al. (2025) are tailored for goal-reaching tasks and are therefore excluded from our comparison; we discuss them separately in **??**.

- **MISL Baselines:** The most typical DIAYN (Eysenbach et al., 2019a) and the successor feature variant APS (Liu & Abbeel, 2021b).

- **Pure Exploration Baselines:** Disagreement (Pathak et al., 2019) and CIC (Laskin et al., 2022).

- **Model-Based Baselines:** UMP (Sekar et al., 2020; Rajeswar et al., 2022) and EUCLID (Yuan et al., 2022).

For a fair comparison independent of lower-level implementations, UMP, EUCLID, and our MIDL all employ TDMPC (Hansen et al., 2022) as the model-based RL backbone. UMP learns a dense dynamic model, while EUCLID and our MIDL learn an ensemble. The *Oracle* is the results of running TDMPC with task rewards for 2 million steps.

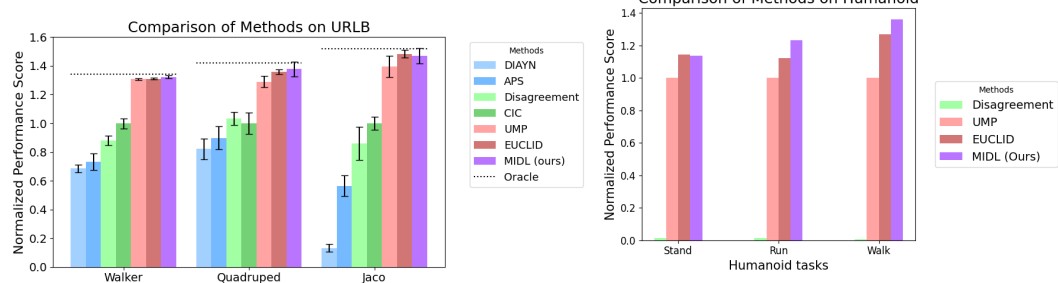

(a) URLB results normalized over CIC. MIDL results are close to the Oracle performance.

(b) Humanoid results normalized over UMP, it displays interquartile means (Agarwal et al., 2021) because Humanoid is sensitive to initial positions.

Figure 5: Comparison of downstream performance across benchmarks.

**Training:** 2 million pretaining steps for all environments, 100,000 finetuning steps for URLB environments, and 150,000 steps for humanoid. 5 independent runs for each downstream task. More details can be found in Section H.

**Result Analysis:** As shown in Tables 3 and 4, model-based methods (UMP, EUCLID, MIDL) consistently outperform the strongest model-free baselines (CIC/Disagreement). On standard URLB tasks, MIDL slightly exceeds the state-of-the-art ensemble method EUCLID, achieving a total score of 8072 compared to EUCLID's 8058, both approaching oracle performance. On more complex, high-dimensional humanoid tasks, the model-free method Disagreement performs poorly, while MIDL establishes a clear advantage over other model-based methods, achieving a 4% higher interquartile mean than EUCLID.

Table 4: Results for humanoid.

| Task | Disagreement | UMP | EUCLID | MIDL |
|---|---|---|---|---|
| stand | 4±2 | 298±18 | **339±22** | **338±3** |
| run | 1±1 | 82±7 | 90±6 | **101±11** |
| walk | 2±1 | 261±22 | 337±29 | **355±47** |

## 5 CONCLUSION

We identified a major limitation in current URL research: mainstream MISL methods often fail to learn useful skills for realistic tasks. To address this, we advocate for a paradigm shift from unsupervised skill learning to unsupervised dynamic learning. For this purpose, we propose MIDL, a principled approach designed to efficiently learn diverse and accurate dynamic ensembles and enhance downstream task performance. By focusing on dynamic learning, MIDL not only addresses the shortcomings of existing methods but also opens new avenues for advancing the field of URL. Furthermore, MIDL provides insights into how world models can be effectively learned and leveraged for real-world applications, inspiring future research in this direction.

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

## A   LIMITATIONS

While our method provides a insight on the new direction of URL and demonstrates strong performance across various tasks, there are several limitations worth noting:

Each model in the MIDL ensemble is trained on data corresponding to its designated skill. Skills learned through a mutual information objective are typically well-separated from one another, meaning that each model is exposed to only a limited region of the data. However, data from other regions could potentially enhance its learning. To address this, we explore alternative data matching strategies in Section G, which allow each model to select suitable data from the combined dataset, thereby improving the learning of dynamics. They offer promising directions for future research in model-based URL.

MIDL requires training and maintaining an ensemble of models together with an auxiliary gating network, which increases computational and memory costs compared to model-free approaches. Although this overhead is manageable in our experiments, scaling to more complex or real-time settings may demand further efficiency improvements. Notably, the ensemble design is analogous to a Mixture of Experts (MoE), where for the same total number of parameters, MoEs often yield lower inference cost and greater robustness.

Our experiments are primarily conducted in simulated environments, including a toy 2D domain and simulated continuous control benchmarks. While these settings are suitable for controlled evaluations of model diversity and pretraining efficacy, real-world deployment may introduce challenges such as noisy observation and partial observability.

We assume a fixed number of models throughout training. Future research could be on how to dynamically adapt the ensemble size for continual learning setting.

## B   RELATED WORKS

### B.1   RELATED UNSUPERVISED SKILL LEARNING METHODS

For behavior-learning URL methods, we have already mentioned both exploration methods and skill learning methods in Section 2, and provided more details in Section D.

Here we discuss other related works such as METRA (Park et al., 2024b), which has recently gained considerable attention. METRA is specifically designed for goal-reaching tasks and, like its predecessors LSD (Park et al., 2022) and CSD (Park et al., 2023), does not support fine-tuning on downstream tasks with general reward functions. Its evaluation has been limited to goal-reaching settings. As state-of-the-art unsupervised skill discovery methods remain constrained to simple locomotion or navigation tasks, this reinforces our central motivation: the need for a new paradigm capable of learning general-purpose dynamics applicable to a wide range of downstream tasks.

A follow-up work, HILP (Park et al., 2024a), addresses this limitation by solving a linear regression problem to select the best latent skill $z$ for downstream tasks. However, it is not a full-stack URL method, since it relies on other URL methods for data collection and does not introduce a fine-tuning mechanism beyond zero-shot skill selection via linear regression. Moreover, as shown in their Table 3, even with the best dataset, HILP underperforms compared to CIC (one of our baselines) and falls significantly short of MIDL and other model-based approaches.

Table 5: Performance comparison for URLB, the values are total scores that sum the scores of all tasks for each environment

| Method | Walker | Quadruped | Jaco |
|--------|--------|-----------|------|
| HILP   | 2912   | 2436      | 236  |
| CIC    | 2943   | 2587      | 599  |
| MIDL   | **3722** | **3518** | **879** |

Although the settings differ slightly, our approach uses 2M pretraining steps and 100k finetuning steps, while HILP uses up to 10M offline transitions and 10k labeled downstream task transitions for

linear regression. Despite having access to more data, HILP still underperforms because it lacks a fine-tuning mechanism that leverages gradient-based updates from downstream task data.

## B.2 MIXTURE OF EXPERTS

The Mixture of Experts (MoE) (Jacobs et al., 1991; Shazeer et al., 2017) is a deep learning structure in which multiple expert models are trained jointly and a gating network dynamically selects a subset of them for each input. This conditional computation allows MoE models to scale the number of parameters without proportionally increasing inference cost, since only a few experts are active at a time. Recent advances in sparse MoEs (Shazeer et al., 2017; Puigcerver et al., 2023; Lepikhin et al., 2021; Fedus et al., 2022) have demonstrated state-of-the-art results across large-scale natural language processing and vision tasks, offering improved efficiency, robustness, and specialization of experts. The ensemble structure of MoE closely aligns with our proposed MIDL framework: each dynamics model can be viewed as an expert specialized in a distinct region of the state space, while the gating network ensures efficient coordination. This connection suggests that principles from MoE research, such as sparse activation and expert specialization, can be leveraged to further improve the scalability and efficiency of URL methods.

## B.3 WORLD MODELS

World models aim to learn a predictive representation of the environment, capturing its dynamics in a compact latent space. By simulating transitions and outcomes internally, they enable sample-efficient reinforcement learning, planning, and exploration without interacting directly with the environment. Classical model-based RL approaches (Ha & Schmidhuber, 2018; Hafner et al., 2020b; Chua et al., 2018) focus on learning dynamics from low- to medium-dimensional continuous environments, often using stochastic latent models or ensembles to improve robustness and uncertainty estimation. More recently, the concept of world models has been extended to leverage large pretrained language models (LLMs) (Wang et al., 2024; Schick & et al., 2023), treating text-based reasoning or code-executable simulations as a form of predictive environment modeling. Such LLM-based world models can capture complex, high-level, and multi-modal dynamics, suggesting a pathway for general-purpose unsupervised pretraining. Regardless of whether the representations are states, text, or images, our proposed MIDL provides a principled approach to explore and learn a Mixture-of-Experts world model that is both robust and accurate.

## C BROADER IMPACT

This work advances the field of unsupervised reinforcement learning (URL) by proposing a method (MIDL) that improves model-based pretraining through learning diverse dynamics. It contributes to building more sample-efficient, generalizable agents by enabling better world models in environments without external rewards. This could reduce the reliance on expensive human supervision and reward design, benefiting applications in robotics, simulation, and embodied AI.

## D UNSUPERVISED RL BASELINES

### D.1 MUTUAL INFORMATION SKILL LEARNING METHODS

Some literature (Laskin et al., 2021) refers to this category as competence-based methods. However, there are other Wasserstein distance skill learning methods (He et al., 2022; Yang et al., 2024) that also learn skills and fall under the "competence-based" category. To emphasize their specific objective in skill learning, we classify them as Mutual Information Skill Learning (MISL) methods.

**DIAYN** (Eysenbach et al., 2019b): Diversity is All you need (DIAYN) is a typical mutual information skill learning method that maximizes the mutual information between states and skills $I(Z; S) = H(Z) - H(Z|S)$. $H(Z)$ is kept maximal by sampling $z$ from a discrete uniform prior distribution. The $-H(Z|S)$ is estimated with a discriminator $\log q(z|s)$

$$r_t^{\text{DIAYN}} \propto \log q(z_t|s_t) + \text{ const.} \tag{11}$$

**APS** (Liu & Abbeel, 2021b): Active Pretraining with Successor Features (APS) combines the particle-based entropy estimation in Singh et al. (2003) and the successor feature formulation in Hansen et al. (2020), which optimizes

$$r_t^{\text{APS}} \propto \sum_{j \in \text{random}} \log \|s_t - s_j\| + \log q(z_t|s_t), \; j = 1, ..., K, \tag{12}$$

where K nearest neighbors are used to estimate the entropy for a given state or image embedding, and $\log q(z_t|s_t)$ is represented by a successor feature formulation $\phi^T(s^t)z_t$ for fast task adaptation.

### D.2 PURE EXPLORATION METHODS

We used Disagreeement (Pathak et al., 2019) and the entropy maximization version of CIC (Laskin et al., 2022) as the baselines for pure exploration methods, without skill learning. They are the best-performing model-free URL methods for the URL benchmark (Laskin et al., 2021).

**Disagreement** (Pathak et al., 2019): Disagreement considers the variance of the predictions of forward model ensemble $\{g_i (\mathbf{s}' \mid \mathbf{s}, \mathbf{a}), i = 1, ..., N\}$ as the intrinsic reward to encourage the agent to explore uncertain regions. It approximately optimizes the information gain $I(S', \Theta|S, A)$ (Mazzaglia et al., 2022; Sekar et al., 2020).

$$r_t^{\text{Disagreement}} \propto \text{Var} \{g_i (\mathbf{s}_{t+1} \mid \mathbf{s}_t, \mathbf{a}_t)\} \quad i = 1, \ldots, N. \tag{13}$$

**CIC** (Laskin et al., 2022): Contrastive Intrinsic Control (CIC) employs noise contrastive estimation (Gutmann & Hyvärinen, 2010) to learn a representation of state transitions $\tau = (s, s')$, with the objective of optimizing $H(\tau)$. While their paper claims to maximize $I(\tau; Z)$ as the objective, the intrinsic reward of their implementation only optimizes $H(\tau)$. Its intrinsic reward is

$$r_t^{\text{CIC}} \propto \sum_{j \in \text{random}} \log \|g(\tau_t) - g(\tau_j)\|, \; j = 1, ..., K, \tag{14}$$

where $g(\tau)$ is the state transition representation learned by noise contrastive estimation.

### D.3 MODEL-BASED METHODS

**UMP**: Model-based Pretraining (UMP) is a baseline we implemented to represent the model-based URL methods that focus on learning a single dynamic model using pure exploration techniques. Methods such as those proposed by Sekar et al. (2020) and Rajeswar et al. (2022) fall into this category, as their intrinsic reward is based on exploratory information gain, $I(S', \Theta|S, A)$, estimated using Disagreement, or $I(S', Z'|S, A)$, estimated using LBS (Mazzaglia et al., 2022). For the UMP baseline implemented in Section 4, we chose Disagreement as the exploration method to isolate the effects of different exploration strategies across UMP, EUClID, and our proposed MIDL.

$$r_t^{\text{UMP}} \propto \text{Var} \{g_i (\mathbf{s}_{t+1} \mid \mathbf{s}_t, \mathbf{a}_t)\} \quad i = 1, \ldots, N. \tag{15}$$

**EUCLID**: EUCLID (Yuan et al., 2023) learns an ensemble of dynamic models. Similar to our MIDL, the $i$-th model in the ensemble is learned by the data collected by its specific "skill" $z^{(i)}$. They encourage the diversity of skill by regulating with the KL-divergence between conditional policies. Their intrinsic reward is:

$$r_t^{\text{UMP}} \propto \text{Var} \{g_i (\mathbf{s}_{t+1} \mid \mathbf{s}_t, \mathbf{a}_t)\} + \log \frac{\tilde{\pi}(s_t|z_t)}{\pi(s_t|z_t)} \quad i = 1, \ldots, N, \tag{16}$$

where $\tilde{\pi} = \frac{1}{N} \sum_i^N \pi(s|z^{(i)})$ is the average policy.

### D.4 CATEGORIZATION OF URL METHODS

By the terminology in Laskin et al. (2021; 2022), CIC is hard to categorize strictly into one of the established categories—knowledge-based, competence-based, or data-based. Its intrinsic reward does not explicitly optimize $I(\tau; Z)$ but its representation learning needs to optimize the NCE loss which maximizes the similarity between state transitions and corresponding skills, thus optimizing a form

of mutual information $I(\tau; Z)$. Additionally, The estimation of its entropy $H(\tau)$ depends on not only raw data but also on the representation function $g$,, blending elements of both data-driven and knowledge-based approaches.

In our opinion, the existing categories place too much emphasis on lower-level implementation details, such as whether methods are parametric or nonparametric, rather than focusing on their underlying objectives. For instance, SMM (Lee et al., 2019) and ProtoRL (Yarats et al., 2021)both aim to maximize state entropy but employ different lower-level approximations, leading to their classification into different categories despite their shared goal. This highlights a limitation in the current categorization framework.

A more intuitive and meaningful approach would be to categorize URL methods based on their primary objectives. For example:

- **Entropy-based methods**: SMM, ProtoRL, CIC (focus on maximizing state or transition entropy),

- **Information gain-based methods**: Disagreement, LBS (focus on maximizing information gain about dynamics or latent variables),

- **Mutual Information Skill Learning methods**: DIAYN, APS (focus on learning skills by maximizing mutual information between skills and states or trajectories).

This objective-driven categorization would provide clearer distinctions between methods and better reflect their core intentions, making it easier to understand and compare different approaches in the field.

## E    PRACTICAL APPROXIMATION DETAILS

### E.1    DERIVATION FROM (4) TO (5)

The term $p(\hat{s}' \mid s, a, s', z)$ in (4) is intractable, so we approximate it using a Gaussian distribution

$$q(\hat{s}'|s, a, s', z) := \mathcal{N}(\hat{s}' \mid s', \Sigma),$$

following a standard variational approximation. In our framework, this Gaussian $q(\hat{s}' \mid s, a, s', z)$ corresponds directly to the model prediction $q_\theta(s' \mid s, a, z)$ in (5). This equivalence arises because the distribution of $q_\theta(s' \mid s, a, z)$ is $\mathcal{N}(s' \mid \hat{s}', \Sigma)$, which is identical to $\mathcal{N}(\hat{s}' \mid s', \Sigma)$. Therefore, we have

$$q_\theta(s' \mid s, a, z) = q(\hat{s}'|s, a, s', z) \approx p(\hat{s}'|s, a, s', z).$$

### E.2    ESTIMATING THE INFORMATION GAIN $I(\hat{S}'; S'|S, A)$:

We can expand the information gain to have

$$I(S'; \hat{S}'|S, A) = H(S'|S, A) - H(S'|S, A, \hat{S}') \tag{17}$$

This is commonly implemented based on ensemble disagreement (Shyam et al., 2019; Sekar et al., 2020). Since the ensemble members are often conditional Gaussians with means as outputs of the neural networks and fixed variance, they consider that $p(S'|S, A, \hat{S}')$ has a constant variance regardless of $\hat{S}'$. The marginal entropy $H(S'|S, A)$ is maximized when the ensemble means are far apart (disagreement) with the least overlaps, then their probability mass is maximally spread out.

By treating $p(\hat{S}')$ as a mixture distribution of parameter point masses:

$$p(\Theta) := \frac{1}{|\mathcal{Z}|} \sum_{z \in \mathcal{Z}} \delta(\hat{S}' - \hat{s}'_z) \tag{18}$$

where $\theta_z$ is the $z$th-model parameter of the ensemble.

We, like Sekar et al. (2020); Park & Levine (2023), use the empirical variance (disagreement) over ensemble means to measure how far apart they are

$$\text{Dis}(s,a) = \frac{1}{|\mathcal{Z}| - 1} \sum_{z \in \mathcal{Z}} (\mu_\theta(s,a,z) - \mu_\theta(s,a)) \tag{19}$$

where $\mu_\theta(s,a,z)$ is the mean of the predicted next state distribution and $\mu_\theta(s,a) = \frac{1}{|\mathcal{Z}|} \sum_{z \in \mathcal{Z}} \mu_\theta(s,a,z)$

As discussed in Section 3, the information gain term can be estimated via prediction error, similar to ICM (Pathak et al., 2017). This approximation is more principled in the Gaussian case, where $\log p(s'|s,a,\hat{s}')$ aligns closely with the prediction error $||s' - \hat{s}'||$ between the true next state $S'$ and the predicted next state $\hat{S}'$. Additionally, entropy $H(S'|S,A)$ captures aleatoric uncertainty, which can remain high even if the model is well trained, so there is no need to explicitly estimate and maximize $H(S'|S,A)$.

## F    MIXTURE OF DYNAMICS

In previous literature (Sharma et al., 2020; Park & Levine, 2023), it was common to approximate $p(s'|s,a)$ by

$$q_\theta(s'|s,a) \approx \sum_{z \in \mathcal{Z}} \frac{1}{|\mathcal{Z}|} q_\theta(s'|s,a,z). \tag{20}$$

However, this is a very coarse approximation, and this approximated $q_\theta(s'|s,a)$ can be inaccurate for downstream task adaptation. Here we propose to represent the mixture of dynamics $p(s'|s,a)$ by:

$$q_{\theta,\psi}(s'|s,a) = \sum_z q_\theta(s'|s,a,z) q_\psi(z|s,a) \tag{21}$$

We learn an additional prediction model $q_\psi(z|s,a)$ into the ensemble, which is used as the gating network of the ensemble. During pretraining, it is learned to minimize the prediction error of $q_{\theta,\psi}(s'|s,a)$ while fixing $\theta$ for downstream task fine-tuning, it can be used in a zero-shot manner to facilitate ensemble prediction for planning.

### F.1    APPROXIMATION ERRORS

The approximation of the marginal probability, such as $p(s'|s,a)$ or $p(s'|s)$ has not been very rigorous in previous model-based RL literature. In Sharma et al. (2020); Park & Levine (2023), they originally approximate $p(s'|s)$ by

$$q_\theta(s'|s) \approx \sum_{z \in \mathcal{Z}} \frac{1}{|\mathcal{Z}|} q_\theta(s'|s,z) \tag{22}$$

which represents a coarse approximation with a uniform prior $p(z|s,a) = \frac{1}{\mathcal{Z}}$.

Moreover, their pretraining objective consists of the log probability of $q_\theta(s'|s)$ but their code implementation estimates $\log q_\theta(s'|s)$ by another inaccurate approximation:

$$\log q_\theta(s'|s) \approx \sum_{z \in \mathcal{Z}} \frac{1}{|\mathcal{Z}|} \log q_\theta(s'|s,z) \leq \log \sum_{z \in \mathcal{Z}} \frac{1}{|\mathcal{Z}|} q_\theta(s'|s) \tag{23}$$

where the inequality introduces an additional layer of imprecision additional to the coarse approximation in Equation (22). This compounding of approximations may lead to further deviations from the true objective, highlighting the need for more refined methods to achieve accurate results.

As mentioned, we aim to approximate $p(s'|s,a)$. Instead of using a uniform prior $q_\theta(s'|s,a,z) \approx \sum_{z \in \mathcal{Z}} \frac{1}{|\mathcal{Z}|} q_\theta(s'|s,a,z)$ we propose to represent the mixture of dynamics by:

$$q_{\theta,\psi}(s'|s,a) = \sum_z q_\theta(s'|s,a,z) q_\psi(z|s,a) \tag{24}$$

where $q_\psi(z|s,a)$ serves as a better approximation for $p(z|s,a)$ compared to the uniform prior $\frac{1}{\mathcal{Z}}$, This refinement helps mitigate the first layer of compounding error introduced by coarse approximations of Equation (22) or Equation (20).

The second layer of compounding error in log probability approximation can be avoided by using the $logsumexp$ function of Pytorch (Paszke et al., 2017):

$$\log q_{\theta,\psi}(s'|s,a) = \log \sum_z e^{\log q_\theta(s'|s,a,z) + \log q_\psi(z|s,a)} = logsumexp\left(\log q_\theta(s'|s,a,z) + \log q_\psi(z|s,a)\right)$$

(25)

## F.2 COMPUTING LEARNING OBJECTIVES

It is common practice to model the dynamics for continuous control tasks as a diagonal Gaussian distribution with fixed variance. Under this assumption, the expressions in Equations (9) and (10) can be straightforwardly approximated using the L2 prediction error of the dynamic models.

For learning the $z$-th model $\theta_z$, the mean of $q_\theta(s'|s,a,z)$ is the model output $f_{\theta_z}(s,a)$ and maximizing Equation (9) is equivalent to

$$\min_\theta \mathbb{E}_z \left[ \mathbb{E}_{s,a,s' \sim p(D|z)} \|f_{\theta_z}(s,a) - s'\|^2 \right].$$

(26)

For Equation (10), we do not need $logsumexp$ function for exact computation because we can optimize its lower-bound

$$\max_\psi \mathbb{E}_{s,a,s' \sim D} \left[ \sum_z q_\psi(z|s,a) \log q_\theta(s'|s,a,z) \right] \leq \mathbb{E}_{s,a,s' \sim D} \left[ \log \sum_z q_\theta(s'|s,a,z) q_\psi(z|s,a) \right].$$

(27)

Replacing $\log q_\theta(s'|s,a,z)$ by $-\|f_{\theta_z}(s,a) - s'\|^2$, we get

$$\min_\psi \mathbb{E}_{s,a,s' \sim D} \left[ \sum_z q_\psi(z|s,a) \|f_{\theta_z}(s,a) - s'\|^2 \right].$$

(28)

For the robotic control experiments in Section 4, the transition dynamics can be regarded as single-modal or nearly deterministic. Therefore, we implemented this approach for the experiments. In environments with more stochastic transition dynamics, we can instead apply the more principled approaches outlined below.

## G ALTERNATIVE DYNAMIC MATCHING APPROACHES

### G.1 EXPECTATION-MAXIMIZATION APPROACH

We can also use the data collected by other latent variables $z'$ to train model $\theta_z$. By treating the latent variable $z$ as unknown, we can treat the ensemble as a Gaussian mixture distribution and optimize it with the EM algorithm (Dempster et al., 1977).

In the E-step, we optimize:

$$\min_{\psi_k} \mathbb{E}_{s,a,s' \sim D} \left[ D_{\mathrm{KL}}(q_{\psi_k}(z|s,a) \| p(z|s,a,s',\theta)) \right],$$

(29)

where $p(z|s,a,s',\theta)$ can be estimated by

$$p(z|s,a,s',\theta) = \frac{q_\theta(s'|s,a,z) q_{\psi_{k-1}}(z|s,a)}{q_{\theta,\psi_{k-1}}(s'|s,a)}$$

(30)

In the M-step, we optimize $\theta$ by maximizing the lower bound

$$\max_\theta \mathbb{E}_{s,a,s' \sim D} \left[ \sum_z p(z|s,a,s',\theta) \log q_\theta(s'|s,a,z) \right].$$

(31)

The EM approach can better handle the stochastic dynamics than the naive approach introduced in Section F.2, particularly in multi-modal scenarios, as demonstrated in Figure 6.

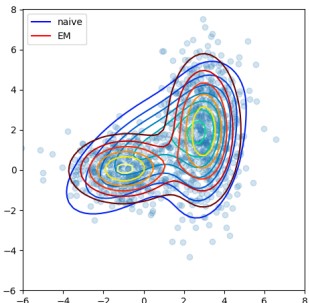

Figure 6: The EM approach is better than the naive approach to handle multi-modal cases

### G.2   OPTIMAL TRANSPORT DYNAMIC MATCHING

The model ensemble $\theta$ can be learned by:

$$\max_{\theta} \mathbb{E}_{s,a,s'\sim D}[\log q_\theta^{ot}(s'|s,a))] \tag{32}$$

Where

$$q_\theta^{ot}(s'|s,a) = \sum_z q_\theta(s'|s,a,z)p^{ot}(z|s,a) \tag{33}$$

During training, we reassign $p^{ot}(z|s,a)$ by solving:

$$\inf_{\gamma\in\Gamma(B(s,a),p(z))} \mathbb{E}_{(s,a,s')\sim B}[-\log q_\theta^\gamma(s'|s,a)] = \mathbb{E}_{(s,a,s')\sim B}[-\log \sum_z q_\theta(s'|s,a,z)\frac{\gamma(s,a,z)}{B(s,a)}] \tag{34}$$

Where $B$ is the batch of data uniform sampled from the combined buffer of all $D_z$, $B(s,a)$ can be approximated as $\frac{1}{|B|}$, and $\Gamma(u,v)$ is the set of all joint probabilities whose marginals are $u$ and $v$. $p^{ot}(z|s,a)$ can be obtained by

$$p^{ot}(z|s,a) = \frac{\gamma(s,a,z)}{B(s,a)} \tag{35}$$

Because:

$$\mathbb{E}_{(s,a,s')\sim B}[-\log \sum_z q_\theta(s'|s,a,z)\frac{\gamma(s,a,z)}{B(s,a)}] \leq \mathbb{E}_{(s,a)\sim B}\mathbb{E}_{s'\sim B(s'|s,a)}\mathbb{E}_{z\sim \frac{\gamma(s,a,z)}{B(s,a)}}[-\log q_\theta(s'|s,a,z)] \tag{36}$$

$$= \mathbb{E}_{(s,a,z)\sim\gamma}\left[\mathbb{E}_{s'\sim B(s'|s,a)}[-\log q_\theta(s'|s,a,z)]\right] \tag{37}$$

The inequality of Equation (36) is tight when some $p^{ot}(z|s,a) = 1$, so the objective in Equation (34) is equivalent to:

$$\inf_{\gamma\in\Gamma(B(s,a),p(z))} \mathbb{E}_{(s,a,z)\sim\gamma}\left[\mathbb{E}_{s'\sim B(s'|s,a)}[-\log q_\theta(s'|s,a,z)]\right] \tag{38}$$

After the reassignment, model parameter $\theta$ is updated by minimizing the objective of Equation (38). The intuition behind this reassignment procedure of samples is that no matter how the transition samples are collected, we can obtain a joint distribution $\gamma(s,a,z)$ such that cross entropy (prediction error) is minimized while satisfying the marginal constraints. Minimized prediction error after assignment means that models are assigned to their most suitable regions of the explored state space. The marginal constraints of uniform distributions $B(s,a)$ and $p(z)$ ensure that all samples are chosen with equal importance for model update and that all models in the ensemble are updated by equally weighted losses. This avoids situations when one less accurate model is ignored in training or when hard samples are never chosen for the model update.

## H  TRAINING DETAILS

### H.1  TRAINING SETTINGS

In the pre-training stage, all components of the model are trained in an unsupervised fashion, using samples gathered from 2 million steps of interaction in a reward-free environment. Afterward, the fine-tuning stage begins, where each agent undergoes fine-tuning for specific downstream tasks using extrinsic rewards, with the training duration set to 100,000 steps. However, for the humanoid domain, this is extended to 150,000 steps to accommodate its increased complexity. To ensure the reliability of our results, we base our analysis on 5 independent runs for each downstream task. For consistency, DDPG (Lillicrap et al., 2016) serves as the backbone for all model-free baselines and for the pretraining model-based methods. The experimental setup follows the same configuration as in URLB (Laskin et al., 2021). The model-based methods employ Model Predictive Path Integral (MPPI) (Williams et al., 2015) for the planning in the finetuning stage with a horizon of 5. The number of models in the ensemble is 4 by default. To ensure a fair comparison, we maintain consistent training settings across all methods.

### H.2  DETAILED RESULTS OF THE TOY EXAMPLE

The learnable parameter for each model in the dynamics is the wind vector, constrained to have the same magnitude $(0.2\sqrt{2})$ as the wind. This table shows the wind learned by different methods.

Table 6: Learned wind vectors in each model of the ensemble across methods. Only MIDL successfully captures all four distinct quadrant-specific dynamics.

|  | UMP | EUCLID | MIDL (Ours) |
|---|---|---|---|
| Model 1 | (0.2003, -0.1983) | (0.1991, -0.1999) | (0.2003, -0.1991) |
| Model 2 | (0.0675, -0.2738) | (0.1997, 0.1997) | (-0.1998, 0.1993) |
| Model 3 | (-0.2003, -0.1990) | (-0.2000, 0.1993) | (-0.1994, -0.1999) |
| Model 4 | (-0.1996, -0.1996) | (-0.1976, 0.2017) | (0.1991, 0.1996) |

Since each quadrant has a distinct wind vector: $[0.2 \operatorname{sign}(x), 0.2 \operatorname{sign}(y)]$, only MIDL successfully discovers all four distinct quadrant-specific wind vectors.

### H.3  PREVIOUS RESULTS

Disagreement result is from Laskin et al. (2021). UMP and EUCLID results are from Yuan et al. (2023).

### H.4  COMPUTATIONAL RESOURCES

All experiments are run on a single A100 GPU with 72 CPU cores. Every process takes 2GB of memory. The estimated running time for a single run is shown in Table 7.

Table 7: Estimated computational time (in hours) for pretraining and finetuning across different domains.

| Domain | Pretraining Time (hrs) | Finetuning Time (hrs) |
|---|---|---|
| Walker | 8.0 | 1.0 |
| Quadruped | 10.0 | 1.5 |
| Jaco | 16.0 | 2.5 |
| Humanoid | 30.0 | 4.5 |

Table 8: Hyper-parameters of the exploration algorithms used in our experiments.

| DIAYN hyper-parameter | Value |
|---|---|
| Skill dim | 16 |
| Skill sampling frequency | 50 |
| Discriminator net | $512 \rightarrow 1024$ |
| | $\rightarrow 1024 \rightarrow 16$ ReLU MLP |

| CIC hyper-parameter | Value |
|---|---|
| Skill dim | 64 continuous |
| Prior | Uniform [0,1] |
| Skill sampling frequency | 50 |
| Skill net arch. $g_{\psi_1}$ | $(|\mathcal{O}|) \rightarrow 1024$ |
| | $\rightarrow 1024 \rightarrow 64$ ReLU MLP |
| Skill net arch. $g_{\psi_2}$ | $64 \rightarrow 1024$ |
| | $\rightarrow 1024 \rightarrow 64$ ReLU MLP |
| Prediction net arch. | $64 \rightarrow 1024$ |
| | $\rightarrow 1024 \rightarrow 64$ ReLU MLP |

| APS hyper-parameter | Value |
|---|---|
| Representation dim | 512 |
| Reward transition | $\log(r + 1.0)$ |
| Successor feature dim. | 10 |
| Successor feature net arch. | $(|\mathcal{O}||) \rightarrow 1024$ |
| | $\rightarrow 1024 \rightarrow |\mathcal{O}|$ ReLU MLP |
| $k$ in NN | 12 |
| Avg top $k$ in NN | True |
| Least square batch size | 4096 |

| Disagreement hyper-parameter | Value |
|---|---|
| Ensemble size | 5 |
| Forward net | $(|\mathcal{O}| + |\mathcal{A}|) \rightarrow 1024$ |
| | $\rightarrow 1024 \rightarrow |\mathcal{O}|$ ReLU MLP |

## H.5 HYPER-PARAMETERS

For a fair comparison, we standardized the model-based backbone to TDMPC (Hansen et al., 2022). As a result, most of the dynamic model parameters remain consistent with the original TOLD implementation in TDMPC (Hansen et al., 2022), and some of them change during the pre-training phase (PT) and the fine-tuning phase (FT). Following prior work (Hafner et al., 2020a; Hansen et al., 2022), we use a task-specific action repeat hyperparameter for the URLB benchmark based on DMControl. This hyperparameter is set to 2 by default, except for the Quadruped domain, where it is set to 4. EUCLID employs a multi-head dynamic model to represent the ensemble and introduces a new policy head every 500k steps. We adhered to this implementation to ensure that the only variable being compared is the intrinsic reward design, isolating its impact on performance.

Table 9: Hyper-parameters of the dynamic model, policy, and planner.

| Dynamic model | Value |
| --- | --- |
| Batch size | 1024 |
| Max buffer size | 1e6 |
| Latent dim | 50 (default) |
| | 100 (Humanoid) |
| MLP hidden dim | 256 (Encoder) |
| | 1024 (otherwise) |
| MLP activation | ELU |
| Optimizer ($\theta$) | Adam |
| Learning rate | 1e-4 (PT) |
| | 1e-3 (FT) |
| Reward loss coefficient ($c_1$) | 0.5 |
| Consistency loss coefficient ($c_2$) | 2 |
| Value loss coefficient ($c_3$) | 0.1 |
| $\theta^-$ update frequency | 2 |
| **Policy** | **Value** |
| Seed steps | 0 (PT) |
| | 4000 (FT) |
| Discount factor ($\gamma$) | 0.99 |
| Action repeat | 2 (default) |
| | 4 (Quadruped) |
| **Planning (Only for FT phase)** | **Value** |
| Iteration | 6 |
| Planning horizon ($L$) | 5 |
| CEM population size | 512 |
| CEM elite fraction | 12 |
| CEM policy fraction (Policy/CEM) | 0.05 |
| CEM Temperature | 0.5 |
| **Multi-head** | **Value** |
| Num of prediction heads ($H$) | 4 |
| specific interval time steps $\mathcal{T}$ | 500k |

