# OpenReview forum: "Mutual Information Dynamics Learning: A New Paradigm for Unsupervised Reinforcement Learning"
_ICLR.cc/2026/Conference — Submitted to ICLR 2026_

### Official Review · Reviewer_u9Md · 2025-10-28

**Soundness:** 2
**Presentation:** 3
**Contribution:** 2
**Rating:** 4
**Confidence:** 4

**Summary:**

The paper introduces a new unsupervised reinforcement learning (URL) objective based on mutual information between the next state predicted by a dynamics model and the real next state, with a skill latent conditioned on the current state and action. The objective encourages a skill-conditioned policy to cover diverse dynamics while being distinct. An ensemble of models is trained from the data collected by the policy, where each model is specialized for one corresponding skill.
At finetuning the pre-trained policy and ensemble dynamics models used for planning, and then the policy is fine-tuned on the data collected by the planning scheme.
The method is evaluated on the URL benchmark (URLB) environment and on the humanoid environment.

**Strengths:**

- The paper is mostly clear and easy to follow.
- Results are strong and promising.

**Weaknesses:**

- Experiments are mainly focusing on locomotion environments (only one manipulation task), this is might be sufficient for locomotion tasks, but I think more evaluation is needed on manipulation benchmarks.
- The derivation of the objective is not clear, especially the jump from equations 4 to 5, and the justification does not make sense.

**Questions:**

1. I find it confusing that, in the pre-training phase, there is an additional policy trained with a pure exploration term. How can we guarantee that the data collected by this policy will be accurate or useful for training one of the specialized models? How does this policy affect the theory introduced in the previous section? Also, since Equation (8) already includes a pure exploration term, in principle, there should be no need for an additional pure exploration policy. Did the authors run their method with a policy trained only using Equation (8)? How does that affect the performance of the method during both pre-training and fine-tuning? Would removing the pure exploration policy make a significant difference?
1. Can you clarify the choice of $z$ in the model during fine-tuning? Do you use the selector network $\phi$? I suggest adding an additional line in the pseudocode in Algorithm 2 to clarify this step.
1. Have you evaluated the exploration performance of the pre-trained policies? For example, by counting the number of covered states or (x,y) positions (when feasible)? Do you think the coverage would be correlate with fine-tuning performance?
1. Can you also include a plot or table showing performance before and after fine-tuning, or a learning curve for the fine-tuning process? I think this is important to assess the efficiency of the fine-tuning procedure.
1. The results on URLB are not conclusive (the confidence bounds overlap).
1. Can you clarify how many runs were performed for the experiments in Section 4.3? Line 464 says “5 independent runs for each downstream task.” Does that mean that pre-training was done once? Is each of the five runs initialized with a different random seed? Also, can you add the confidence bounds to Figure 5b?
1. In Appendix E.1, I do not understand the justification for using $q_{\theta}(s' \mid s, a, z)$ in place of $q(\hat{s}' \mid s, a, s', z)$. Can you explain this argument more clearly?

I am willing to raise my score if the authors addressed most of my concerns.

---

> ### Author Response · Authors · 2025-11-18
> **Rebuttal (1/2)**
>
> Thank you for your detailed review and valuable feedback. We appreciate the opportunity to address your questions and hope to clarify any points that may improve your understanding of our work.
>
> ---
>
> > ## Q1: About Additional Pure Exploration Policy.
>
> **A1:** This pure exploration policy $z'$ is not for training specialized models, but rather for **downstream task initialization**, helping to explore high-reward regions of the downstream task. Effective initialization for downstream task adaptation requires policies with broad state-space coverage, so pure exploration policies are more effective than specialized policies. This also explains why skill-learning methods often perform worse than pure exploration methods, as shown in Figure 3.
>
> In our method, specialized policies is only used to collect specialized data for model training, while pure exploration is used for downstream task initialization, combining the advantages of both approaches.
>
> Removing the pure exploration policy would force the use of a specialized policy for downstream initialization. Since specialized policies typically cover the state space less broadly, convergence on sparse-reward downstream tasks would likely be slower.
>
> > ## Q2: About Choice of $z$ during finetuning
>
> **A2:** Yes, the selector network $\phi$ is indeed used during downstream-task planning. We will revise Algorithm 2 to clearly reflect this usage.
>
> > ## Q3: About Evaluation for Exploration
>
> **A3:** Yes, we have evaluated the particle-based state entropy [1]. For model-free methods, we observe a clear correlation between state-space coverage and downstream fine-tuning performance, especially on sparse-reward tasks such as Jaco.
>
> Although the overall state entropy of the skill-learning method DIAYN is comparable to that of the pure-exploration method Disagreement (both around ~1400, with the exact number depending on the entropy estimation hyperparameters), each individual skill learned by DIAYN covers only a small portion of the state space of approximately ~300 in entropy. As a result, when a downstream task has sparse rewards, a DIAYN skill is unlikely to explore or reach the high-reward region, making fine-tuning substantially harder.
>
> |method|average performance|state entropy|
> |-|-|-|
> |DIAYN|19.5|$\approx$ 300 (each skill)|
> |Disagreement|128.3|$\approx$ 1400|
>
> > ## Q4: About Learning Curve
>
> **A4**: Yes, we can add this to the appendix in the revision.
>
> Below are examples of learning curves for MIDL during fine-tuning:
> |Time|Walker-Run|Walker-Flip|Jaco-ReachBottomLeft|Humanoid-Stand|Humanoid-Run|
> |-|-|-|-|-|-|
> |0|96|290|0|17|4|
> |10k|144|343|8|10|1|
> |20k|575|694|88|30|1|
> |30k|691|933|112|89|1|
> |40k|700|953|90|83|44|
> |50k|745|957|114|188|59|
> |60k|764|963|190|247|66|
> |70k|721|972|179|82|80|
> |80k|763|980|223|243|80|
> |90k|781|976|193|299|97|
> |100k|786|978|219|304|91|
> |110k|-|-|-|324|96|
> |120k|-|-|-|335|94|
> |130k|-|-|-|330|10|
> |140k|-|-|-|317|110|
> |150k|-|-|-|341|112|
>
> For simpler environments such as Walker, the zero-shot performance is already non-zero. However, for sparse tasks like Jaco or more challenging tasks such as Humanoid-Run, the zero-shot performance is very low and requires more learning steps before improvement becomes visible. If the initial policy is not sufficiently exploratory, this early learning phase can be slow, which in turn affects fine-tuning efficiency.
>
> For learning curves of the model-free baseline methods, please refer to Appendix D of the URLB paper.

---

> > ### Comment · Reviewer_u9Md · 2025-11-24
> > **Response to authors**
> >
> > Thank you for your response and the clarifications.
> >
> > > This pure exploration policy is not for training specialized models, but rather for downstream task initialization,
> >
> > This is not clear from Algorithm 1, line 329 (or line 5 in Algorithm 1) says to use $\pi_{\phi}^z$ to collect data $D_Z$ and then if $z$==${z'}$ it updates the policy with the pure exploration reward.
> >
> > > We will revise Algorithm 2 to clearly reflect this usage
> >
> > Have you updated the paper? Can you upload the revised version? if you are using the selector network to choose the skill z, how are you using pure exploration policy for initialization? What if the selector did not choose $z'$ at the start of fine-tuning?  This part is confusing.
> >
> > Best.

---

> > > ### Author Response · Authors · 2025-11-24
> > > **Further response**
> > >
> > > Thanks very much for the continued discussion. Due to a technical issue with the main author's email and Overleaf account, we may need to upload the full revised manuscript slightly later. In the meantime, we provide the revised versions of **Algorithm 1** and **Algorithm 2** here for clarity.
> > >
> > > > ## Regarding the remaining unclear point about Algorithm 1
> > >
> > > The clarified version of **Algorithm 1** is as follows:
> > >
> > > > **Algorithm 1** Pretraining of MIDL
> > > > 1. **Input:** number of models in ensemble $|\mathcal{Z}|$
> > > > 2. **Initialize:** Model $\theta$, Selector $\psi$, Policy $\pi_\phi$, Dataset $D=[D_1,D_2,...,D_{|\mathcal{Z}|}]$
> > > > 3. **While** pretraining **do**
> > > > 4. Sample $z$ uniformly from $\mathcal{Z} ∪ z′$
> > > > 5.  **If** $z \neq z'$ **then:**
> > > > 6. Rollout with $\pi_\phi^{z}$ and collect data to $D_z$
> > > > 7. Update $\theta,\psi$ with Equations (9) and (10)
> > > > 8. Update $\pi_\phi^z$ with MIDL intrinsic reward with Equation (8)
> > > > 9. **else:**
> > > > 10. Update $\pi_\phi^{z'}$ with pure exploration intrinsic reward
> > > > 11. **End If**
> > > > 12. **End while**
> > >
> > > We hope this resolves the remaining confusion regarding the pretraining procedure.
> > >
> > > > ## Regarding the remaining unclear point about Algorithm 2
> > >
> > > The selector network is **not** used for policy selection; rather, it selects the most suitable dynamics model from the ensemble to support model-based planning. During finetuning, the acting policy used to initialize planning is always $\pi_\phi^{z'}$.
> > >
> > > At each step, the planning module simulates several steps ahead using the ensemble of learned dynamics models, and the selector identifies the most appropriate model from the ensemble to guide this simulation.
> > >
> > > The clarified **Algorithm 2** is:
> > >
> > >
> > > > **Algorithm 2** Finetuning of MIDL
> > > > 1. **Input:** Model $\theta$, Selector $\psi$, Policy $\pi_\phi$.
> > > > 2. **Initialize:** Dataset $D$.
> > > > 3. **While** finetuning **do**
> > > > 4. Planning with $\pi_\phi^{z'}$ and model ensemble $\theta,\psi$, and collect data to $D$
> > > > 5. Update $\theta,\psi$ with data from $D$
> > > > 6. Update $\pi_\phi^{z'}$ with data from $D$ with normal RL updates
> > > >7. **End While**
> > >
> > > We hope this clarifies the finetuning stage and resolves the remaining questions.

---

> > > > ### Comment · Reviewer_u9Md · 2025-11-26
> > > >
> > > > Dear authors,
> > > >
> > > > Thank you for your response and the clarification. And I think the algorithm makes sense.
> > > > I still think it should be evaluated across more environments, as I think URLB is limited (to all locomotion environments except one).
> > > >
> > > > Would it be possible to run the method on Crafter, MinGrid, or some Atari games and report how it performs?
> > > >
> > > > Best.

---

> ### Author Response · Authors · 2025-11-18
> **Rebuttal (2/2)**
>
> > ## Q5: About Overlap in URLB
>
> **A5:** One reason the performance gains of our method over EUCLID appear relatively small on URLB is that both MIDL and EUCLID already achieve performance **close to the oracle** level (i.e., the agent trained directly with task reward), as indicated by the dotted horizontal line in Figure 5(a). In contrast, for the more complex Humanoid tasks, where the performance ceiling is higher, the improvement achieved by MIDL is more substantial.
>
> Beyond downstream performance, we also provide evidence in Section 4.2 showing that the data collected by MIDL is more challenging and that the resulting learned dynamics model is more **accurate** and **robust** than that of EUCLID. We further include a new model-accuracy analysis for the Ant-v3 environment, which highlights structural limitations of EUCLID. The payoff matrix below summarizes model prediction errors:
>
>
> |Data\model|MIDL(ours)|EUCLID|UMP|Difficulty$\uparrow$|
> |-|-|-|-|-|
> |MIDL|6.13|106.00|7.88|**120.01**|
> |EUCLID|0.79|0.06|0.27|1.12|
> |UMP|6.04|106.25|5.23|117.52|
> |Accuracy$\downarrow$|**12.96**|212.31|13.38||
>
> For this case, results show that MIDL both (1) collects the **most challenging and diverse data** and (2) learns the **most accurate dynamics model**. In contrast, EUCLID tends to **overfit** to its own data and struggles to generalize. We observe that EUCLID’s collected data exhibits **low state entropy** (522 in the table below, calculated by particle-based entropy [1]), likely due to its action-level KL regularization. This KL term can lead to a local optimum where skills maximize action diversity without substantially improving state-space exploration. For example, each skill may collapse to using only a single distinct action.
>
> |Method|Particle-based Entropy|
> |-|-|
> |MIDL|13604|
> |Euclid|522|
> |UMP|19410|
>
> > ## Q6: About Runs for Pretraining and Finetuning
>
> **A6:** Since pre-training requires significantly more computation, we pre-train the model three times and select the run with the highest state entropy for fine-tuning. For each selected pre-trained model, we then perform five independent downstream runs, each initialized with a different random seed.
>
> Figure 5b reports the interquantile mean, which excludes the best and worst seeds and averages over the middle 50%. We use this metric because the Humanoid environment is highly sensitive to initialization, and poor initial states can cause irreversible failures (e.g., the humanoid falls and never gets back up). The results from all seeds, along with standard deviations, are reported in Table 4.
>
> > ## Q7: About Derivation from (4) to (5)
>
> We appreciate the careful inspection. Since $p(\hat{s}' \mid s, a, s', z)$ is intractable, we adopt a standard variational approximation and model it using a Gaussian distribution:
>
> $$
> p(\hat{s}' \mid s, a, s', z) \approx \mathcal{N}(\hat{s}' \mid s', \Sigma)
> = \frac{1}{\sqrt{(2\pi)^d |\Sigma|}}
> \exp\Bigg(-\frac{1}{2} (\hat{s}' - s')^\top \Sigma^{-1} (\hat{s}' - s') \Bigg),
> $$
>
> where $d$ is the dimension of the state $\hat{s}'$, and $\Sigma$ is the covariance matrix.
>
> By symmetry of the Gaussian, we can switch $\hat{s}'$ and $s'$, and equivalently write:
>
> $$
>  \frac{1}{\sqrt{(2\pi)^d |\Sigma|}}
> \exp\Bigg(-\frac{1}{2} (\hat{s}' - s')^\top \Sigma^{-1} (\hat{s}' - s') \Bigg)= \frac{1}{\sqrt{(2\pi)^d |\Sigma|}}
> \exp\Bigg(-\frac{1}{2} (s'-\hat{s}')^\top \Sigma^{-1} (s'-\hat{s}') \Bigg)=\mathcal{N}(s' \mid \hat{s}', \Sigma)
> $$
>
> There, we have
>
> $$p(\hat{s}' \mid s, a, s', z) \approx \mathcal{N}(\hat{s}' \mid s', \Sigma)=\mathcal{N}(s' \mid \hat{s}', \Sigma)$$
>
> Since $\mathcal{N}(s' \mid \hat{s}', \Sigma)$ corresponds exactly to the output of our forward dynamics models, we can use these models to estimate $p(\hat{s}' \mid s, a, s', z)$.
>
> ---
>
> ## Reference
>
> We thank the reviewers again for their valuable feedback and hope our responses address all the concerns raised.
>
> [1] Singh et al. Nearest neighbor estimates of entropy.

---

> ### Author Response · Authors · 2025-11-27
>
> Thank you for the thoughtful feedback and for the improved score.
>
> Crafter, MiniGrid, and most Atari games primarily involve discrete action spaces. The MIDL is a general framework, and it should be possible to integrate it with suitable backbones like Muzero. Investigating this direction would be interesting and is a promising avenue for future research.

---

### Official Review · Reviewer_DfsZ · 2025-10-29

**Soundness:** 2
**Presentation:** 2
**Contribution:** 3
**Rating:** 2
**Confidence:** 3

**Summary:**

This paper proposes a new paradigm, MIDL, which shifts from unsupervised skill learning to unsupervised dynamic learning, addressing the limitations of MISL approaches in capturing complex behaviors.

**Strengths:**

In the MIDL framework, a gating network is trained instead of assuming a uniform prior over skills. This network may better samples appropriate skills for a given state–action pair.

**Weaknesses:**

It is debatable whether the claim that skill learning should be shifted to dynamics learning under the unsupervised RL paradigm is well supported, as skill learning and model-based learning focus on different aspects of reinforcement learning.

The paper does not clearly explain what $Z$ denotes in the MIDL framework.

In the experiments section, the analysis of the downstream task is insufficient, and some references are missing. Moreover, there is no explanation provided for Fig. 5.

**Questions:**

Could you conduct an additional experiment using ICM [1]?

[1] Deepak Pathak, Pulkit Agrawal, Alexei A Efros, and Trevor Darrell. Curiosity-driven exploration by self-supervised prediction. In International Conference on Machine Learning, 2017.

---

> ### Author Response · Authors · 2025-11-18
> **Rebuttal**
>
> Thank you for your review and feedback. We appreciate the opportunity to answer your questions and clarify the points that may not have been fully conveyed in the original submission.
>
> ---
> > ## About Comparison with ICM
>
> The ICM results on the URL benchmark are listed below:
>
> | Domain        | Task               | Score    |
> | ------------- | ------------------ | -------- |
> | **Walker**    | Flip               | 514 ± 25 |
> |               | Run                | 388 ± 30 |
> |               | Stand              | 913 ± 12 |
> |               | Walk               | 713 ± 31 |
> | **Quadruped** | Jump               | 205 ± 33 |
> |               | Run                | 133 ± 20 |
> |               | Stand              | 329 ± 58 |
> |               | Walk               | 143 ± 31 |
> | **Jaco**      | Reach bottom left  | 72 ± 22  |
> |               | Reach bottom right | 117 ± 18 |
> |               | Reach top left     | 116 ± 22 |
> |               | Reach top right    | 94 ± 18  |
>
>
> Overall, ICM achieves only **0.65×** the total score of our baseline method, *Disagreement*.
>
> For the more challenging humanoid environments, ICM, like the other model-free exploration method such as *Disagreement*, shows **nearly zero** performance, making it not comparable to our model-based methods.
>
> Since both ICM and Disagreement are pure exploration methods, they are expected to exhibit comparable behavior. Therefore, we chose *Disagreement* as a more representative and better-performing baseline for comparison.
>
> ---
>
> > ## About Motivation for the Shift to Dynamics Learning
>
> On the standard URL benchmarks (Figure 3), skill-learning methods often perform **worse** than pure exploration approaches **without** skill learning, indicating that the learned skills tend to be simple and not transferable. Even the most recent skill-learning approaches [1,2] have only been evaluated on relatively simple goal-reaching tasks, highlighting the need for an alternative direction. This motivates our shift toward dynamics learning.
>
> > ## About Notation $Z$
>
> $Z$ denotes the index of a dynamic model in the ensemble, and $\pi(\cdot|\cdot,z)$ is the policy used to collect data for training the $z$th dynamic model. This is mentioned in section 3.1, lines 242-249.
>
> > ## About Result Analysis
>
> The results in Figure 5 are discussed in lines 467–475.
> Figure 5(a) shows downstream task performance on the URL benchmark, and Figure 5(b) shows results for the Humanoid environment. In these plots:
>
>
> * **Blue bars** represent skill-learning methods,
>
> * **Green bars** represent pure exploration methods,
>
> * **Pink–purple bars**  represent model-based methods.
>
> The comparison shows that skill-learning methods perform worse than pure exploration methods, while model-based methods achieve the best performance overall.
>
> We agree that the current analysis could be expanded with better clarity, and we will provide a more detailed discussion and add the missing references in the revision.
>
>
> ---
> > ## Reference
>
> We thank the reviewers again for their valuable feedback and hope our responses address all the concerns raised.
>
> [1] Park et al. Controllability-aware unsupervised skill discovery
>
> [2] Park et al. METRA: scalable unsupervised RL with metricaware abstraction.

---

> > ### Comment · Reviewer_DfsZ · 2025-11-27
> >
> > thank the authors for their response; however, most of my concerns remain:
> >
> > **Shift to dynamics learning:** The motivation for shifting skill learning toward dynamics learning is unconvincing, given that previous skill-learning methods aim to solve simple goal-reaching tasks.
> >
> > **Notation of $Z$**: The use of the notation $Z$ continues to be confusing. In Figure 4, $z$ represents a skill; in line 200, the paper states that “our approach promotes skill discovery”; and in line 315, it notes that “each model is learned from the data collected by its corresponding skill.”. These usages appear inconsistent and make it difficult to clearly understand the intended meaning.
> >
> > **Experimental analysis**: Lines 467–475 still refer only to Tables 3 and 4. In both tables, the performance of the model-based methods are largely overlapping. Similarly, in Figure 5(a), the results for MIDL and EUCLID overlap. Yet, no deeper analysis is provided to explain this behavior.
> >
> > Finally, the authors did not take advantage of the option to update the PDF, which would have allowed them to show concretely how these concerns were addressed.

---

> ### Author Response · Authors · 2025-11-27
>
> Thank you for your follow-up response. Due to a technical issue with the main author's email and Overleaf account, we may need to upload the fully revised manuscript slightly later. In the meantime, we address your remaining concerns below.
>
> **Shift to dynamics learning:**
>
> Exactly because skill-learning methods are inherently designed for goal-reaching tasks and therefore perform well only in that narrow setting, and because they can even perform worse than pure exploration methods without skill learning on common ULRB benchmarks, it is necessary to shift the community’s focus from skill learning to learning knowledge that can generalize to broader and more complex downstream tasks, such as the knowledge of the underlying environmental dynamics. We mentioned this in our previous response, and this point is also discussed in lines 176–203 of the original manuscript. This motivation was also clear to the other reviewers.
>
> **Notation of $Z$**:
>
> The notation of $Z$ refers to the latent variable that conditions the policy used to collect data for training the models in the ensemble. Each $z$ can be viewed as a “skill” that collects data to train the corresponding $z$-th model in the ensemble. The phrase *“promotes skill discovery”* in line 200 is followed by
>
> *“promotes skill discovery that generates diverse state transitions, enabling the training of an ensemble of specialized dynamics models.”*
>
> Therefore, the use of $Z$ in Figure 4 and in lines 200 and 315 of the original manuscript is consistent. In all cases, $Z$ represents the “skills” that collect diverse data to train a diverse and robust model ensemble, and each $z$ indexes the model trained with the data collected by skill $z$. The notation was clear to the other reviewers.
>
> Besides, we plan to provide a clearer presentation of the algorithm, which should help better illustrate the role of $Z$.
>
> The clarified version of **Algorithm 1** is as follows:
>
> > **Algorithm 1** Pretraining of MIDL
> > 1. **Input:** number of models in ensemble $|\mathcal{Z}|$
> > 2. **Initialize:** Model $\theta$, Selector $\psi$, Policy $\pi_\phi$, Dataset $D=[D_1,D_2,...,D_{|\mathcal{Z}|}]$
> > 3. **While** pretraining **do**
> > 4. Sample $z$ uniformly from $\mathcal{Z} ∪ z′$
> > 5.  **If** $z \neq z'$ **then:**
> > 6. Rollout with $\pi_\phi^{z}$ and collect data to $D_z$
> > 7. Update $\theta,\psi$ with Equations (9) and (10)
> > 8. Update $\pi_\phi^z$ with MIDL intrinsic reward with Equation (8)
> > 9. **else:**
> > 10. Update $\pi_\phi^{z'}$ with pure exploration intrinsic reward
> > 11. **End If**
> > 12. **End while**
>
> The clarified **Algorithm 2** is:
>
>
> > **Algorithm 2** Finetuning of MIDL
> > 1. **Input:** Model $\theta$, Selector $\psi$, Policy $\pi_\phi$.
> > 2. **Initialize:** Dataset $D$.
> > 3. **While** finetuning **do**
> > 4. Planning with $\pi_\phi^{z'}$ and model ensemble $\theta,\psi$, and collect data to $D$
> > 5. Update $\theta,\psi$ with data from $D$
> > 6. Update $\pi_\phi^{z'}$ with data from $D$ with normal RL updates
> >7. **End While**
>
> **Experimental analysis:**
>
> As explained in lines 472–473, the overlap in Figure 5(a) occurs because both MIDL and EUCLID are **close to the oracle** performance (the dotted horizontal line).
>
> In the original manuscript, we analyzed model accuracy in **Section 4.2** to demonstrate the empirical advantages of MIDL over EUCLID in terms of accuracy and robustness.
>
> The example in **Section 4.1** illustrates the theoretical advantage of having diverse models within the learned ensemble.
>
> We further include a new model-accuracy analysis for the Ant-v3 environment, which highlights structural limitations of EUCLID. The payoff matrix below summarizes model prediction errors:
> |Data\model|MIDL(ours)|EUCLID|UMP|Difficulty$\uparrow$|
> |-|-|-|-|-|
> |MIDL|6.13|106.00|7.88|**120.01**|
> |EUCLID|0.79|0.06|0.27|1.12|
> |UMP|6.04|106.25|5.23|117.52|
> |Accuracy$\downarrow$|**12.96**|212.31|13.38||
>
> For this environment, results show that MIDL both (1) collects the **most challenging and diverse data** and (2) learns the **most accurate and robust** dynamics model. In contrast, EUCLID tends to overfit to its own data and struggles to generalize. We observe that EUCLID’s collected data exhibits low state entropy (522 in the table below, calculated by particle-based entropy [1]), likely due to its action-level KL regularization. This KL term can lead to a local optimum where skills maximize action diversity without substantially improving state-space exploration. For example, each skill may collapse to using only a single distinct action.
> |Method|Particle-based Entropy|
> |-|-|
> |MIDL|13604|
> |EUCLID|522|
> |UMP|19410|
>
>
>
>
>
> ---
> **Reference**
>
> [1] Singh et al. Nearest neighbor estimates of entropy.

---

### Official Review · Reviewer_QGL6 · 2025-11-01

**Soundness:** 2
**Presentation:** 3
**Contribution:** 2
**Rating:** 4
**Confidence:** 4

**Summary:**

Presents a mutual-information objective that combines Mutual Information Skill Learning(MISL) with model-based pretraining to learn a specialized ensemble of dynamics models.

maximize information gain + dynamic matching

Reports that this yields more diverse/accurate dynamics and SOTA downstream performance on the standard Unsupervised Reinforcement Learning Benchmark (URLB) and competitive performance on Humanoid, with compatiblility with standard model-based backbones.

**Strengths:**

Clear motivation citing that mutual information skill learning struggles to transfer beyond simple navigation

Mutual information-based decomposition combines exploration (info gain via prediction variance/error) with model specialization, and the proposed method avoids the uniform-prior approximation used in prior work.

**Weaknesses:**

Performance gains on URLB are not noticeable over EUCLID: The total scores are very close (8072 vs 8058), and the stronger advantage appears mainly on Humanoid, so more analysis across various settings and initializations, etc are needed.

Training an ensemble + gating would increase memory/computation requirements and the authors mention this, but this should be discussed more, e.g. varying ensemble size vs performance vs computation

Derivations rely on Gaussian/variance proxies. What happens under model missspecification?

No ablation study

**Questions:**

Ablation of the three core elements will be needed : dynamic-matching term, information-gain term, and gating vs. uniform mixing. which is most critical for the examples presented?

Humanoid is known to be very sensitive to initial conditions; what happens when tested with more wider distribution across seeds and initial conditions ?

Performance gain over EUCLID seems weak. Can you provide benefit for the tasks where each (or any) of the core elements noticeably better performance?

---

> ### Author Response · Authors · 2025-11-18
> **Rebuttal (1/2)**
>
> Thank you for your detailed and valuable review. We address your questions and concerns below.
>
> ---
>
> >## About Ablation
>
> Because we implemented UMP and EUCLID with the same backbone and dynamic models, they actually serve as ablations.
>
> | Method       | Information-Gain     |  Additional Term for Ensemble Diversity |
> | ------- | -------- | -------- |
> | UMP | Disagreement | None  |
> | EUCLID | Disagreement | Policy diversity |
> | MIDL | Disagreement | Dynamic-matching |
>
> Because exploration is fundamental for unsupervised RL, every method has an information gain term for exploration and they all use the same Diagreement for fair ablation. UMP serves as an ablation of MIDL **without dynamic-matching term** and EUCLID serves as an ablation of MIDL using **an alternative term** for diverse ensemble learning.
>
> For uniform mixing, we observe a decrease in the accuracy of the learned model. Similar to the results in Table 2, the table below compares prediction errors between uniform mixing and gating in the Ant environment:
>
> |Data\model|Gating|Uniform|Difficulty$\uparrow$|
> |-|-|-|-|
> |Gating|6.13|11.34|**17.47**|
> |Uniform|5.64|5.05|10.69|
> |Accuracy$\downarrow$|**11.77**|16.39|-|
>
> > ## About Gains over EUCLID
>
> One reason the performance gains of our method over EUCLID appear relatively small on URLB is that both MIDL and EUCLID already achieve performance **close to the oracle** (i.e., training agent with task reward directly), which is represented by the dotted horizontal line in Figure 5(a). In contrast, for the more complex Humanoid tasks, whose performance ceiling is higher, the improvement achieved by MIDL is more substantial.
>
> Beyond downstream performance, we also provide evidence in Section 4.2 showing that (1) the data collected by MIDL is more challenging, and (2) the resulting learned dynamics model is more accurate than that of EUCLID. We further include a new model-accuracy analysis for the Ant-v3 environment, which highlights structural limitations of EUCLID. The payoff matrix below summarizes model prediction errors:
>
>
> |Data\model|MIDL(ours)|EUCLID|UMP|Difficulty$\uparrow$|
> |-|-|-|-|-|
> |MIDL|6.13|106.00|7.88|**120.01**|
> |EUCLID|0.79|0.06|0.27|1.12|
> |UMP|6.04|106.25|5.23|117.52|
> |Accuracy$\downarrow$|**12.96**|212.31|13.38||
>
> For this environment, results show that MIDL both (1) collects the **most challenging and diverse data** and (2) learns the **most accurate dynamics model**. In contrast, EUCLID tends to **overfit** to its own data and struggles to generalize. We observe that EUCLID’s collected data exhibits **low state entropy**(522 in the table below, calculated by particle-based entropy [1]), likely due to its action-level KL regularization. This KL term can lead to a local optimum where skills maximize action diversity without substantially improving state-space exploration. For example, each skill may collapse to using only a single distinct action.
>
> |Method|Particle-based Entropy|
> |-|-|
> |MIDL|13604|
> |EUCLID|522|
> |UMP|19410|
>
> >##  About Memory/Computation Requirements
>
> The additional memory and computation cost mainly come from inference and training of the gating network. The ensemble of dynamics models itself can be viewed as a single dense network whose members are **sparse** variants, following the standard **Mixture-of-Experts (MoE)** formulation. Therefore, the additional cost remains largely **constant** and does not scale significantly with ensemble size. This architecture is widely used in modern LLMs [2], and when implemented properly, the gating network introduces negligible overhead, while sparse expert activation can even reduce computation.
>
> In our experiments (Tables 2–4), we compare MIDL (MoE) with UMP (dense) under the same **total parameter number** for dynamic models. The results show that the MoE design leads to higher model accuracy and better robustness to unseen data while maintaining equal parameter count. This indicates that, given the same computational and memory budget, the MoE-based dynamics model achieves better generalization than dense alternatives.
>
> For pretraining Humanoid, MIDL (sparse with gating) takes approximately 30 hours, which introduces reasonable additional computation when compared to 24 hours for UMP (dense without gating). Increasing the ensemble size does not significantly increase training time, as it only enlarges the output dimension of the small gating network. At the same time, inference time can decrease slightly, since each expert processes a smaller subset of the activated weights.
>
> Overall, the additional computation and memory cost is **not a practical concern**. The number of experts should be chosen based on whether the sparse experts are sufficiently expressive for prediction and whether using too many experts would make the gating network inaccurate.

---

> ### Author Response · Authors · 2025-11-18
> **Rebuttal (2/2)**
>
> >## About Gaussian proxy
>
> Using Gaussian proxies is common in dynamic model learning not only because they are computationally feasible, but also because the **log probability** of a Gaussian reduces to the **L2 prediction error** of the dynamic model. This makes it both easy to compute and intuitively meaningful as a measure of prediction accuracy. Consequently, most previous methods,such as Disagreement, UMP, and DADS, adopt Gaussian proxies.
>
> When the dynamic model is non-Gaussian, estimating the mutual information (MI) objective in eq.3 becomes significantly more expensive and often impractical, since the prediction error no longer provides a valid estimate of the log probability. In such cases, MI estimation requires computing entropies through sampling and numerical integration, which is computationally intensive.
>
>
> Regarding potential model misspecification, if the gating network assigns a suboptimal expert for certain states, the prediction can indeed be less accurate. However, through training, the gating network learns to assign the most suitable model for the explored state space, resulting in robust average performance.
>
> ---
>
> We thank the reviewers again for their valuable feedback and hope our responses address all the concerns raised.
>
> ## Reference
>
> [1] Singh et al.  Nearest neighbor estimates of entropy.
>
> [2] Cai et al. A Survey on Mixture of Experts in Large Language Models

---

### Official Review · Reviewer_VDMV · 2025-11-01

**Soundness:** 4
**Presentation:** 4
**Contribution:** 3
**Rating:** 8
**Confidence:** 4

**Summary:**

The authors propose Mutual Information Dynamics Learning (MIDL), a novel unsupervised reinforcement learning framework in which different skills capture the dynamics of distinct state-space regions. MIDL leverages mutual information objectives to collect data with diverse dynamics, enabling the training of a mixture of specialized dynamics models. These models provide sufficient coverage for solving a wide range of downstream tasks. The framework consists of a pretraining phase that learns the dynamics model ensemble and a finetuning phase that uses model-based planning for task adaptation.

**Strengths:**

- The paper effectively critiques existing MISL methods using empirical evidence and makes a compelling case for prioritizing approaches focusing on dynamics.
- The approach and objectives are mathematically justified. The mutual information objective (Equation 2) decomposes into information gain and dynamic matching terms, with clear intuition provided.
-  The 2D quadrant wind environment (Figure 2, Table 1) provides an interpretable toy experiment that demonstrates MIDL's ability to learn diverse dynamics where baselines fail.
- Strong results across multiple benchmarks (URLB, Humanoid) with diverse baselines. The model accuracy analysis via a two-player game framework (Table 2) is insightful.

**Weaknesses:**

- While Table 2 provides valuable insights into model learning quality, this analysis is only conducted on the Swimmer environment. It would strengthen the claims to demonstrate whether the superior accuracy of MIDL generalizes across other environments.
- Minor typos and formatting issues:
    - "Diagreement" (line 295)
    - "exlploit" (line 388)
    - incorrect citation formatting throughout (using \citet where \citep would be appropriate, e.g. lines 168 and 180)
    - a missing reference at line 434,
    - $\widehat{S}'$ but should be $S'$ line 260
    - $q_\theta (s'|s,a))$ should be $q_{\theta,\psi} (s'|s,a))$ in Equation 8

**Questions:**

- Did you perform the model accuracy analysis (similar to Table 2) for additional environments beyond Swimmer to demonstrate the generalizability of MIDL's superior dynamics learning? If yes, are observations the same?
- How does the performance evolve as you vary the number of skills $|\mathcal{Z}|$?

---

> ### Author Response · Authors · 2025-11-18
>
> Thank you for the detailed and supportive review. We appreciate your feedback and provide our responses and clarifications below.
>
> ---
>
> >## About Further Model Accuracy Analysis
>
> We conducted a model-accuracy analysis for the Ant-v3 environment. The payoff matrix below summarizes model prediction errors:
>
>
> |Data\model|MIDL(ours)|EUCLID|UMP|Difficulty$\uparrow$|
> |-|-|-|-|-|
> |MIDL|6.13|106.00|7.88|**120.01**|
> |EUCLID|0.79|0.06|0.27|1.12|
> |UMP|6.04|106.25|5.23|117.52|
> |Accuracy$\downarrow$|**12.96**|212.31|13.38||
>
> For this case, results show that MIDL both (1) collects the most challenging and diverse data and (2) learns the most accurate dynamics model. In contrast, EUCLID tends to **overfit** to its own data and struggles to generalize. We observe that EUCLID’s collected data exhibits **low state entropy**(522 in the table below, calculated by particle-based entropy [1]), likely due to its action-level KL regularization. This KL term can lead to a local optimum where skills maximize action diversity without substantially improving state-space exploration. For example, each skill may collapse to using only a single distinct action.
>
> |Method|Particle-based Entropy|
> |-|-|
> |MIDL|13604|
> |Euclid|522|
> |UMP|19410|
>
> ## About Varying the Number of $|\mathcal{Z}|$
>
> There are two ways to vary the number of models $|\mathcal{Z}|$ in the ensemble:
>
> 1. Each dynamic model constant parameter size
> 2. Total ensemble has constant parameter size
>
> If $|\mathcal{Z}|$ is increased while keeping each model’s parameter size constant, the total number of parameters and overall model expressiveness increase, leading to better overall accuracy.
>
> If $|\mathcal{Z}|$ is increased while keeping the total parameter count constant, each individual dynamic model becomes less expressive. Although model diversity increases, the reduced expressiveness of each model can negatively impact its individual accuracy.
>
> For both ways, increasing $|\mathcal{Z}|$ would make the ensemble accuracy more sensitive to the choice  of the  gating nework $\phi$,
>
> In both cases, increasing $|\mathcal{Z}|$ makes the ensemble’s overall performance more sensitive to the design and choice of the gating network $\phi$.
>
> ---
>
> We thank the reviewers again for their valuable feedback and hope our responses address the questions raised.
>
> ## Reference
>
> [1] Singh et al. Nearest neighbor estimates of entropy.

---

### Author Response · Authors · 2025-12-01
**Rebuttal Summary for Area Chair**

We sincerely appreciate the reviewers’ time and constructive insights. The following is a brief overview of how we addressed the concerns, along with the latest clarifications exchanged during the discussion phase.


---
Below is a table summarizing the major review points before the rebuttal.
We mark concerns **resolved by additional results** in $\textcolor{blue}{\text{blue}}$, concerns **due to factual mistakes** in $\textcolor{red}{\text{red}}$, and concerns **stemming from misunderstandings** in $\textcolor{brown}{\text{brown}}$.

|Reviewer (rating / confidence)  | VDMV (8 / 4)  |QGL6 (4 / 4)  |DfsZ (2 / 3)  |u9Md (4$\rightarrow$6 / 4)|
|-|-|-|-|-|
|**Strengths**| clear motivation |clear motivation|gating network outperform uniform prior|clear writing and strong presentation|
|| rigorous mathematical derivation |rigorous mathematical derivation||strong and promising results|
||intuitive illustration ||||
||strong empirical results ||||
|**Weeknesses**| $\textcolor{blue}{\text{limited environments}}$  |$\textcolor{red}{\text{no ablation}}$|$\textcolor{brown}{\text{unclear motivation}}$|less manipulation tasks|
||  | computation cost|$\textcolor{brown}{\text{unclear notation}}$|$\textcolor{brown}{\text{unclear derivation}}$|
|| |$\textcolor{brown}{\text{small gains on URLB}}$ |$\textcolor{brown}{\text{small gains on URLB}}$|$\textcolor{brown}{\text{small gains on URLB}}$|
|**Questions("-" if similar to weeknesses)**|-|-|$\textcolor{blue}{\text{additional comparison with ICM}}$|$\textcolor{blue}{\text{exploration evaluation}}$|
|||||$\textcolor{blue}{\text{learning curve}}$|

---
During the rebuttal, we addressed every weakness and question raised by the reviewers. Key points are summarised below:
* **No ablation:** This was a factual misunderstanding. As explained in both the original manuscript and the rebuttal, because all model-based methods share the same backbone, the baselines, such as UMP and EUCLID, with different objective components naturally serve as ablations.
* **Limited environments:** We added model-accuracy analyses for the higher-dimensional Ant environment and demonstrated clear improvements over baseline methods such as EUCLID, while also exposing key limitations of EUCLID that our proposed MIDL effectively addresses.
* **Small gains over EUCLID on URLB:** Both MIDL and EUCLID approach **oracle-level** performance on URLB, so their results appear close. In all other settings (Sections 4.1–4.2 and the additional accuracy analyses), MIDL shows clear advantages over EUCLID. This explanation was acknowledged by **u9Md**, who subsequently increased their rating to **6**, but was overlooked by **DfsZ** and **QGL6**, who did not engage further.
* **Computation cost:** We clarified that the additional computation cost introduced by MIDL is constant and **does not scale** with the model ensemble size.
* **Unclear derivation:** We provided a clarified version of the algorithm in the rebuttal. This fully addressed the concern and was explicitly acknowledged by **u9Md**, who increased their rating to **6**
* **Additional comparisons and evaluations**: We added comparisons with ICM, provided particle-based entropy results to evaluate exploration, and included learning curves to illustrate the training dynamics.

Many concerns seem to treat our work as an incremental contribution focused solely on performance, yet it does achieve strong results in most settings. More importantly, **our contribution goes beyond empirical gains by presenting a promising direction for future research in unsupervised reinforcement learning**.

---

After the rebuttal, **VDMV** remains a high rating of 8, and **u9Md** increased their rating from 4 to 6, indicating that we have successfully addressed most of the concerns. Regarding reviewers with negative ratings: **QGL6** and **DfsZ**:
* **QGL6** This reviewer **did not engage** during the rebuttal process. Their concerns were not critical and partly stemmed from factual mistakes or misunderstandings. We believe we have thoroughly addressed all of their points, and had they engaged in discussion, we are confident we could have clarified these issues.
* **DfsZ** has **low** confidence and did **not** demonstrate sufficient expertise on several aspects. They **misunderstood key motivations and notations** that were clear to the other reviewers. Their requested ICM is an outdated and less relevant baseline (though we have added ICM results as requested). In addition, the “strength” they highlighted, the gating network, represents only a minor technical component (less than 10% of the contribution), suggesting they did not grasp the main ideas of our work.

We kindly request the AC to consider these points when evaluating the validity of their concerns, taking into account QGL6’s lack of engagement and DfsZ’s low confidence and misunderstandings.

---

### Meta-Review · Area_Chair_hTZk · 2026-01-10

**Summary:**

This paper proposes a novel unsupervised reinforcement learning framework in which different skills capture the dynamics of distinct state-space regions. Most reviewers agreed that the paper is well-motivated.

The main concerns are about the limtied performance gain and missing experiments.

Although the authors made efforts during the rebuttal phase and one reviewer decided to raise the score, the final ratings cannot meet the requirement. I'm inclined to recommend it for a (borderline) rejection.

**Reviewer Concerns:**

Concerns about the ablation study and computation burden may be fairly addressed.

**Reviewer Scores:**

No

---

### Decision · Program_Chairs · 2026-01-26

Reject